# Progression through mitosis promotes PARP inhibitor-induced cytotoxicity in homologous recombination-deficient cancer cells

Pepijn M. Schoonen[1], Francien Talens[1,*], Colin Stok[1,*], Ewa Gogola[2], Anne Margriet Heijink[1], Peter Bouwman[2], Floris Foijer[3], Madalena Tarsounas[4], Sohvi Blatter[5], Jos Jonkers[2], Sven Rottenberg[5] & Marcel A.T.M. van Vugt[1]

Mutations in homologous recombination (HR) genes *BRCA1* and *BRCA2* predispose to tumorigenesis. HR-deficient cancers are hypersensitive to Poly (ADP ribose)-polymerase (PARP) inhibitors, but can acquire resistance and relapse. Mechanistic understanding how PARP inhibition induces cytotoxicity in HR-deficient cancer cells is incomplete. Here we find PARP inhibition to compromise replication fork stability in HR-deficient cancer cells, leading to mitotic DNA damage and consequent chromatin bridges and lagging chromosomes in anaphase, frequently leading to cytokinesis failure, multinucleation and cell death. PARP-inhibitor-induced multinucleated cells fail clonogenic outgrowth, and high percentages of multinucleated cells are found *in vivo* in remnants of PARP inhibitor-treated $Brca2^{-/-};p53^{-/-}$ and $Brca1^{-/-};p53^{-/-}$ mammary mouse tumours, suggesting that mitotic progression promotes PARP-inhibitor-induced cell death. Indeed, enforced mitotic bypass through EMI1 depletion abrogates PARP-inhibitor-induced cytotoxicity. These findings provide insight into the cytotoxic effects of PARP inhibition, and point at combination therapies to potentiate PARP inhibitor treatment of HR-deficient tumours.

[1] Department of Medical Oncology, University Medical Center Groningen, University of Groningen, Hanzeplein 1, 9713 GZ Groningen, The Netherlands. [2] Division of Molecular Pathology and Cancer Genomics Netherlands, The Netherlands Cancer Institute, Plesmanlaan 121, 1066 CX Amsterdam, The Netherlands. [3] European Research Institute for the Biology of Ageing, University of Groningen University Medical Center Groningen, Hanzeplein 1, 9713 GZ Groningen, The Netherlands. [4] The CRUK/MRC Oxford Institute, Old Road Campus Research Building, OX3 7DQ Oxford, UK. [5] Institute of Animal Pathology, Vetsuisse Faculty, University of Bern, Laenggassstrasse 122, 3012 Bern, Switzerland. * These authors contributed equally to this work. Correspondence and requests for materials should be addressed to M.A.T.M.v.V. (email: m.vugt@umcg.nl).

BRCA1 and BRCA2 function in the repair of DNA double-strand breaks (DSBs) through homologous recombination (HR), and ensure the protection of stalled replication forks[1]. Defective HR is thought to underlie the progressive accumulation of genomic aberrations, leading to malignant transformation. Indeed, mutations in *BRCA1* and *BRCA2* predispose to tumorigenesis, most frequently involving breast and ovarian cancer[2–4]. Due to their DNA repair defect, *BRCA1/2* mutant cancer cells are more sensitive to platinum-based chemotherapeutics, as observed in preclinical models and in clinical studies[5–7]. In addition, *BRCA1/2* mutant cancers were found to be selectively sensitive to inhibition of the poly-(ADP)ribose polymerase PARP1 (refs 7–9). Unfortunately, however, *BRCA1/2* mutant cancers can acquire resistance and relapse[10].

Mechanistically, PARP1 promotes the repair of non-toxic single-strand DNA breaks[11], which are converted into potentially toxic DSBs during S-phase[8,9]. These DSBs depend on HR for repair, and hence were suggested to cause cell death in HR-defective cancer cells. However, the number of single-strand DNA breaks were not found to be increased after PARP1 depletion or PARP inhibition[11–13], and the synthetic lethal interaction between PARP inhibition and HR deficiency may therefore involve other mechanisms[14,15]. Indeed, PARP1 and BRCA1/2 were shown to orchestrate the protection and restart of stalled replication forks[16–20]. Analogously, PARP1 activity increases during replication[21], and sensitivity to PARP inhibition in *BRCA2* mutant cancer cells can be rescued by mutations that prevent replication fork degradation[22].

Notably, aberrant replication intermediates may persist in G2-phase, and can even be propagated into mitosis[23–27], and cause mitotic aberrancies[28–30]. Whether DNA lesions induced by PARP inhibition in HR-deficient cells persist into mitosis, and if they affect cell division remains unclear.

Here, we study the mechanisms by which PARP-inhibitor-induced DNA lesions affect mitotic progression. We describe that PARP inhibition compromises replication fork stability and leads to DNA lesions that are transmitted into mitosis. During mitosis, these DNA lesions cause chromatin bridges and lead to cytokinesis failure, multinucleation and cell death. Importantly, our data show that progression through mitosis promotes PARP-inhibitor-induced cell death, since forced mitotic bypass. abrogates PARP-inhibitor-induced cytotoxicity.

## Results

**PARP-inhibitor-induced lesions are transmitted into mitosis**. To explore the consequences of PARP inhibition on mitotic progression in HR-defective cancer cells, we depleted BRCA2 in HeLa cells (Fig. 1a). As expected, treatment with the PARP inhibitor olaparib resulted in selective killing of BRCA2-depleted cells (Fig. 1b). In line with roles for BRCA2 and PARP in facilitating replication fork stability[22], we observed compromised replication fork protection using single DNA fibre analysis upon BRCA2 depletion, which was aggravated upon PARP inhibition (Fig. 1c,d). These findings show that PARP inhibition in BRCA2-deficient cancer cells incrementally interferes with replication fork stability. In line with previous studies showing involvement of Mre11 and PTIP in degradation of stalled replication fork in BRCA2-deficient cells, Mre11 inhibition using mirin or PTIP depletion alleviated the fork protection defects (Supplementary Fig. 1A,B)[20,22].

Defective replication fork stability upon PARP inhibition was further underscored by the increase in FANCD2 foci in interphase cells upon BRCA2 depletion. A significant further increase was observed when BRCA2-depleted cells were treated with PARP inhibitor (Fig. 1e). Surprisingly, the increase in FANCD2 foci was only accompanied by minor increases in the numbers of γ-H2AX foci in interphase, suggesting that replication lesions do not per se result in DNA breaks (Supplementary Fig. 1C).

The observed replication lesions were not resolved before mitotic entry, as increased numbers of FANCD2 foci were observed in BRCA2-depleted mitotic cells (Fig. 1f). Again, the numbers of FANCD2 foci increased further upon PARP inhibitor treatment (Fig. 1f). Of note, in PARP inhibitor-treated, BRCA2-depleted mitotic cells, numbers of γ-H2AX foci were increased similarly to FANCD2 foci (Supplementary Fig. 1D). Combined, these data show that PARP inhibition in BRCA2-defective cells leads to replication intermediates that are transmitted into mitosis.

**PARP inhibition causes mitotic chromatin bridges**. Since persistence of unresolved replication intermediates into mitosis may interfere with proper chromosome segregation, we tested whether PARP inhibition-induced mitotic aberrancies. Whereas PARP inhibition did not affect the percentages of anaphase or telophase cells containing chromatin bridges in control cells, depletion of BRCA2 in HeLa cells led to an increased percentage of cells showing chromatin bridge formation, which remained unresolved up until telophase (14 and 17% in BRCA2-depleted cells versus 2% in control-depleted cells) (Fig. 2a,b), in line with previous observations[31]. Strikingly, the number of BRCA2-depleted cells containing anaphase chromatin bridges markedly increased upon olaparib treatment (59 and 65% in BRCA2-depleted, olaparib-treated cells versus 20% in control-depleted cells treated with olaparib) (Fig. 2a,b). Interestingly, whereas most of the olaparib-induced chromatin bridges were resolved before telophase in control cells, chromatin bridges persisted throughout mitosis in BRCA2-depleted cells (49 and 57% in BRCA2-depleted cells versus 6% in control-depleted cells) (Fig. 2b). Furthermore, PARP inhibition also markedly increased the numbers of lagging chromosomes in BRCA2-depleted cells (58 and 53% in BRCA2-depleted cells versus 8% in control-depleted cells) (Fig. 2b, right panel).

These observations were not specific for BRCA2, as depletion of BRCA1 (Supplementary Fig. 2A,B) or RAD51 (Supplementary Fig. 2C,D) also showed a clear induction of PARP-inhibitor-induced chromatin bridges persisting throughout mitosis. Notably, BRCA1 or RAD51 depletion in HeLa cells also increased the amount of lagging chromosomes upon PARP inhibition, although to a lesser extent in BRCA1-depleted cells when compared to RAD51 or BRCA2-depleted cells (Fig. 2b, right panel and Supplementary Fig. 2B,D, right panels). Similar results were obtained upon depletion of BRCA1 or BRCA2 in BT-549 breast cancer cells treated with olaparib. Specifically, PARP inhibitor sensitivity greatly increased upon doxycycline-induced shRNAs depletion of BRCA1 or BRCA2 (Supplementary Fig. 2E,F). Importantly, the numbers of unresolved chromosome bridges (Fig. 2c), and lagging chromosomes (Supplementary Fig. 2G) increased significantly in BRCA1/2-depleted BT-549 cells upon olaparib treatment.

We next investigated whether permanent genetic inactivation of BRCA1 shows a similar increase in the amount of mitotic chromatin bridges, when compared to acute siRNA-mediated BRCA2/BRCA1 inactivation. Indeed, the human HCC1937 breast cancer cell line, harbouring a *BRCA1* deletion as well as a hypomorphic BRCA1 allele with a 5382insC frameshift mutation, showed increased chromatin bridges in anaphase and telophase upon olaparib treatment (Fig. 2d), as well as lagging chromosomes (Supplementary Fig. 2H). To validate these results using

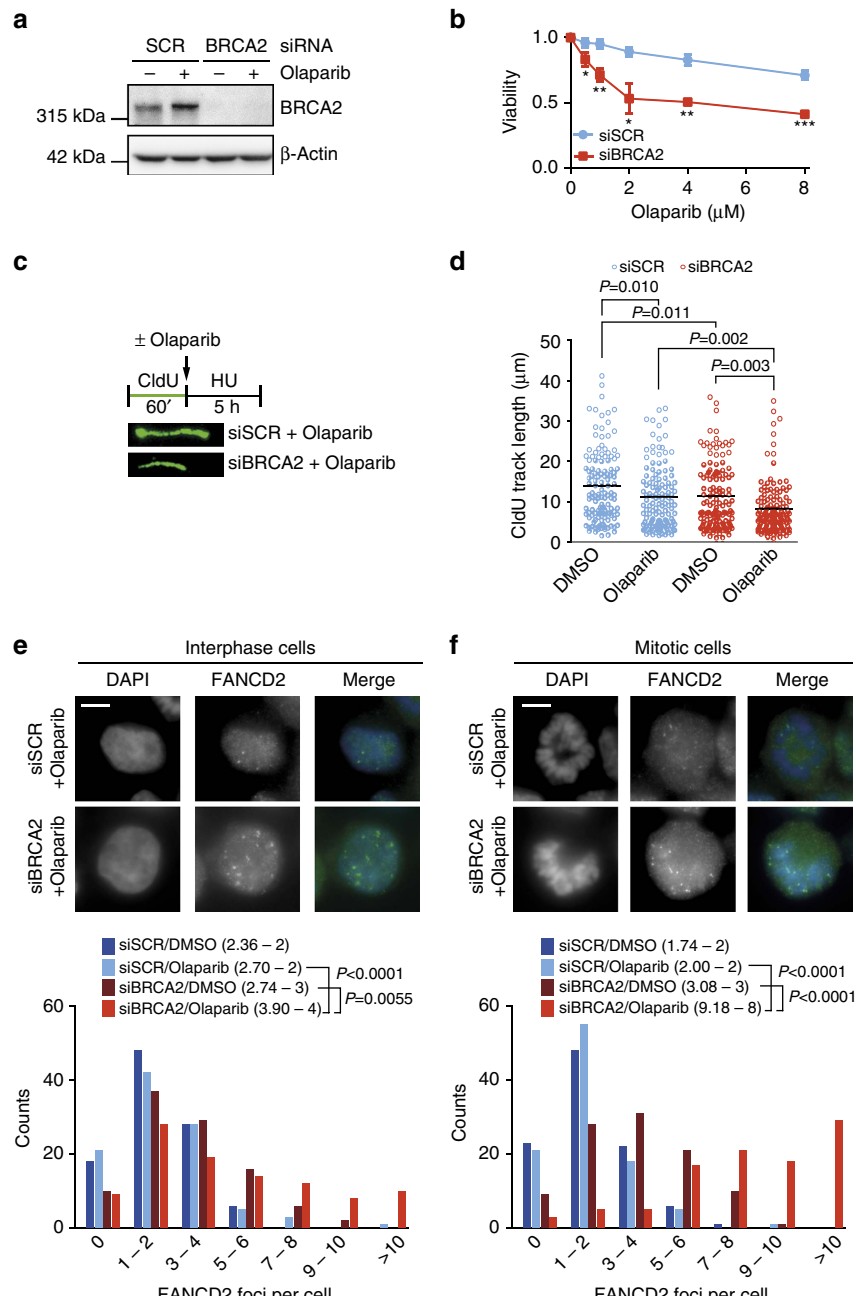

**Figure 1 | PARP-inhibitor-induced lesions are transmitted into mitosis.** (**a**) Immunoblotting of BRCA2 and β-Actin at 48 h after transfection of indicated siRNAs in HeLa cells. Lines next to blots indicate positions of molecular weight markers. (**b**) HeLa cells were transfected with indicated siRNAs for 24 h and subsequently replated and treated with indicated olaparib concentrations for 72 h. Viability was assessed by MTT conversion. Shown graphs are representative of three independent experiments, with three technical replicates each. *P* values were calculated using two-tailed Student's *t*-test. 'NS' indicates not significant. * indicates *P* < 0.05, ** indicates *P* < 0.01, *** indicates *P* < 0.001. (**c**) HeLa cells were transfected with indicated siRNAs and labelled with CldU as indicated. Cells were then treated with HU (5 mM) and DMSO or olaparib (0.5 μM) for 5 h. The DNA was spread into single fibres and CldU track length was determined (125 fibres per condition). (**d**) Quantification of fibre lengths described in **c**. *P* values were calculated using two-tailed Mann–Whitney test. (**e,f**) HeLa cells were transfected with siRNA targeting BRCA2 and treated with DMSO or olaparib (0.5 μM) for 24 h. Cells were stained for FANCD2 (green) and counterstained with DAPI (blue) and the number of FANCD2 foci per nuclei were quantified for interphase cells (**e**) and mitotic cells (**f**). Per condition *n* = 100 nuclei were analysed. Indicated numbers between brackets represent (average—median) from three independent experiments. *P* values were calculated using two-tailed Mann–Whitney test. Throughout the figure 'NS' indicates not significant. All error bars indicate s.d. of three independent experiments.

isogenic models, we next used a tumour cell line derived from a *K14cre;Brca2^{del/del};p53^{del/del}* mouse mammary tumour (denoted as *Brca2^{−/−}*)[32]. As a control, we used an isogenic cell line in which BRCA2 was reconstituted using an infectious bacterial artificial chromosome (iBAC), harbouring the mouse *Brca2* gene

(denoted as *Brca2^{iBAC}*)[33]. Brca2^{iBAC} expression functionally restored BRCA2 function as judged by irradiation-induced foci formation of RAD51 (Supplementary Fig. 3A), and a rescue from PARP inhibitor sensitivity (Supplementary Fig. 3B).

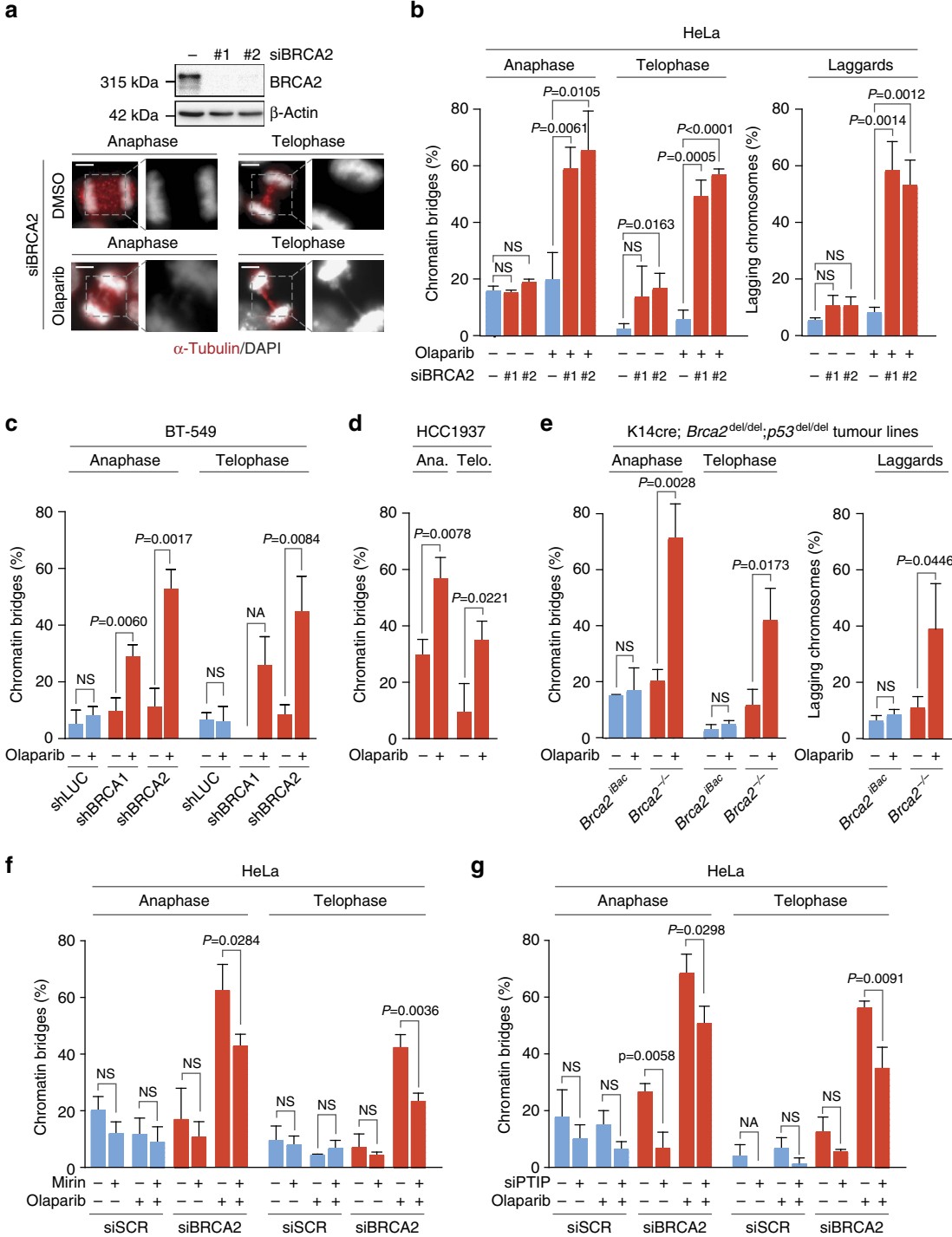

**Figure 2 | PARP inhibition causes mitotic chromatin bridges.** Throughout the figure blue bars represent BRCA1/2 proficient cells, red bars represent BRCA1/2 deficient cells, and P values were calculated using two-tailed Student's t-test. (**a**) HeLa cells were transfected with indicated siRNAs and immunoblotted for BRCA2 and β-Actin levels after 48 h. Lines next to blots indicate positions of molecular weight markers. In parallel, cells were treated with DMSO or olaparib (0.5 μM) for 24 h and stained for α-Tubulin (red) and counterstained with DAPI (white). Representative immunofluorescence images are presented. Scale bars represent 5 μm. (**b**) HeLa cells were treated as for **a**. The percentages of cells containing chromatin bridges (n > 20 events per condition) and lagging chromosomes (n > 40 events per condition) were quantified. (**c**) BT-549 cells harbouring indicated shRNA vectors were pre-treated with doxycycline for 48 h and treated with olaparib (0.5 μM) or DMSO for 24 h. Percentages of cells containing chromatin bridges in anaphase and telophase (n > 20 events per condition) were quantified. (**d**) HCC1937 cells were treated with olaparib (0.5 μM) or DMSO for 24 h. Percentages of anaphase (ana.) or telophase (telo.) cells containing chromatin bridges (n > 20 events per condition) were quantified. (**e**) KB2P1.21 (Brca2$^{-/-}$) and KB2P1.21R1 (Brca2$^{iBac}$) cells were treated with olaparib (0.5 μM) or DMSO for 24 h. Percentages of anaphase or telophase cells containing chromatin bridges (left panel, n > 20 events per condition) and cells containing lagging chromosomes (right panel, n > 40 events per condition) were quantified. (**f,g**) HeLa cells were transfected with indicated siRNAs and after 24 h were treated with olaparib (0.5 μM) or DMSO and/or mirin (50 μM) for 24 h. The percentages of anaphase or telophase cells containing chromatin bridges (n > 20 events per condition) were quantified. Throughout the figure 'NS' indicates not significant and 'NA' indicates not analysable. All error bars indicate s.d. of three independent experiments.

In accordance with our observations after transient BRCA2 depletion, PARP inhibition in $Brca2^{-/-}$ cells resulted in an increased percentage of cells harbouring chromatin bridges in anaphase (71% in $Brca2^{-/-}$ cells versus 17% in $Brca2^{iBAC}$ cells), which to a large degree remained unresolved until telophase (42% in $Brca2^{-/-}$ cells versus 5% in $Brca2^{iBAC}$ cells) (Fig. 2e, left panel). Again, also the percentage of cells with lagging chromosomes was increased (39% versus 8% in $Brca2^{-/-}$ and $Brca2^{iBAC}$ cells, respectively) (Fig. 2e, right panel). These observations again likely reflected generic consequences of defective HR, as very similar defects were observed in a tumour cell line derived from $K14cre;Brca1^{F5-13/F5-13};p53^{F2-10/F2-10}$ mice (Supplementary Fig. 3C). These observations were further validated in DLD-1 human colorectal adenocarcinoma cells, in which $BRCA2$ was inactivated using CRISPR-Cas9. In complete accordance to what was observed in mouse $Brca2$-null cells, $BRCA2^{-/-}$ DLD-1 cells were sensitive to PARP inhibition (Supplementary Fig. 3D), and PARP inhibition greatly enhanced the formation of chromatin bridges as well as lagging chromosomes in $BRCA2^{-/-}$, but not in $BRCA2^{+/+}$ DLD-1 cells (Supplementary Fig. 3E). Combined, these findings indicate that PARP inhibition induces mitotic defects when HR is inactivated acutely or permanently, in a species-independent fashion.

Since the formation of mitotic chromatin bridges and lagging chromosomes in HR-deficient cancer cells are a likely consequence of disturbed replication fork integrity, we tested the involvement of Mre11 and PTIP. Mre11 inhibition or PTIP depletion, which alleviated the PARP-inhibitor-induced replication fork instability (Supplementary Fig. 1A,B), also reduced the amounts of mitotic chromatin bridges (Fig. 2f,g). These findings further corroborate that aberrant control of replication fork stability underlies PARP-inhibitor-induced chromosome bridge formation in mitosis. Notably, the amount of lagging chromosomes was not reduced upon Mre11 or PTIP inactivation, suggesting different biological origins of these lesion (Supplementary Fig. 3F,G).

**PARP trapping is required for chromatin bridge formation**. Cytotoxicity of PARP inhibitors was previously associated with the ability of PARP inhibitor to trap PARP onto DNA, rather than its effects on PARP catalytic activity[15]. To test whether PARP trapping is required for the observed mitotic defects, we depleted PARP1 using siRNA (Fig. 3a). In contrast to treatment with olaparib, a PARP inhibitor which has trapping activity, depletion of PARP1 did not significantly induce mitotic chromatin bridges, nor lagging chromosomes in BRCA2-depleted cells (Fig. 3b and Supplementary Fig. 4A). Also, very similar levels of mitotic chromatin bridges and lagging chromosomes were observed in response to the structurally unrelated PARP inhibitor AZD2461 (Fig. 3c,d and Supplementary Fig. 4B). Of note, the observed increase in chromosome bridges in PARP inhibitor-treated cells was much more pronounced when compared to cisplatin treatment, at a dose that efficiently caused DNA breaks as judged by γ-H2AX (Fig. 3c,d).

Since PARP is involved in multiple cellular processes, we next tested whether the PARP-inhibitor-induced mitotic defects required treatment during S-phase. To this end, cells were synchronized at the G1/S-phase of the cell cycle using a double-thymidine block (Fig. 3e). Cells treated during S-phase displayed significantly increased numbers of mitotic cells with FANCD2 foci (Fig. 3f), chromosome bridges as well as lagging chromosomes (Fig. 3g). In line with expectations, BRCA2-depleted cells with chromosome bridges also contained higher numbers of mitotic FANCD2 foci, when compared to cells without chromosome bridges (Supplementary Fig. 4C). Importantly, when cells

were treated with olaparib past S-phase (at 7 h after release from thymidine block), the number of FANCD2 foci in mitotic cells (Fig. 3f), chromosome bridges as well as lagging chromosomes (Fig. 3g) was significantly reduced. Taken together, these data show that PARP trapping during S-phase is required for induction of mitotic chromosome bridges.

**Chromatin bridges cause multinucleation and cell death**. Unresolved chromatin bridges can cause genomic aberrations, multinucleation and cell death. To investigate the consequences of PARP-inhibitor-induced chromatin bridges for HR-deficient cells, live-cell imaging was used in combination with stable expression of fluorescently tagged Histone-H2B to visualize chromosome dynamics (Fig. 4a). Although some chromatin bridge events were observed in DMSO-treated control cells, the majority of mitoses proceeded either without any visible chromatin bridges (cells with chromatin bridge: 24%) or with chromatin bridges that were resolved during the course of mitosis (6%) (Fig. 4b). Very comparable results were found for control-depleted cells treated with PARP inhibitor (chromatin bridge: 32%; resolved bridge: 14%), or BRCA2-depleted cells treated with DMSO (chromatin bridge: 26%; resolved bridge: 12%) (Fig. 4b). In stark contrast, the amount of aberrant mitoses was strongly increased in BRCA2-depleted cells treated with PARP inhibitor (chromatin bridge: 64%; resolved bridge: 13%) (Fig. 4b and Supplementary Fig. 5A). Furthermore, in a large fraction of BRCA2-depleted cells, unresolved mitotic chromatin bridges resulted in failed cytokinesis leading to multinucleation (29% of total mitoses), or were followed by cell death, (22% of total mitoses), indicating that mitotic failure following PARP inhibitor treatment is often detrimental for BRCA2-deficient cells (Fig. 4b and Supplementary Fig. 5A).

Similar phenotypes were observed in $Brca2$-null mouse mammary tumour cells, expressing GFP-tagged Histone-H2B and mCherry-tagged α-Tubulin (Fig. 4c). Again, PARP inhibitor treatment of $Brca2$-null mouse cells greatly enhanced the number of cells with chromatin bridges (82% versus 22% in $Brca2^{-/-}$ and $Brca2^{iBAC}$, respectively, Fig. 4c). Furthermore, most of the chromatin bridges in $Brca2^{-/-}$ cells were not resolved during mitosis and lead to cytokinesis failure, accompanied with multinucleation (32% versus 5% in $Brca2^{-/-}$ and $Brca2^{iBAC}$, respectively), or cell death (16% versus 1% in $Brca2^{-/-}$ and $Brca2^{iBAC}$, respectively) (Fig. 4c). These effects were not caused by expression of GFP-HistoneH2B or mCherry-α-Tubulin, as similar amounts of cells with >4n DNA content were observed using flow cytometry in BRCA2-depleted HeLa cells or $Brca1/2$-null cells lacking these reporters (Fig. 4d–f, Supplementary Fig. 5B–D). In conclusion, PARP inhibition in cells with inactivated BRCA2 leads to chromatin bridges, which frequently remain unresolved and are associated with multinucleation and cell death.

As PARP inhibitor treatment resulted in a large fraction of multinucleated cells upon cytokinesis failure, we wondered to what extent these cells contribute to clonogenic survival. We therefore sorted BRCA2-depleted cells based on DNA content and separately plated cells with 2n, 4n and >4n DNA content (Fig. 5a and Supplementary Fig. 6A–C). As expected, DMSO-treated, BRCA2-depleted HeLa cells with either 2n or 4n DNA content resulted in efficient clonogenic outgrowth (Fig. 5b). Similarly, 2n or 4n $Brca2^{-/-}$ cells showed comparable numbers of colonies (Fig. 5c). Notably, DMSO-treated cells with >4n DNA content showed a ∼50% decrease in clonogenic potential, indicating that multinucleation reduces viability, but does not per se preclude long-term survival of tumour cells, in line with previous reports[34,35]. Indeed, multinucleated cells did not

display an intrinsic inability to replicate, as judged by BrdU incorporation (Supplementary Fig. 6D,E). Importantly, upon PARP inhibition, clonogenic survival was markedly decreased in both BRCA2-depleted HeLa cells as well as $Brca2^{-/-}$ cells, with 4n DNA-containing cells consistently showing a more

pronounced decrease in clonogenic outgrowth (Fig. 5b,c). Notably, cells with >4n DNA content showed a near-complete loss of colony formation, showing that PARP-induced multinucleation precludes long-term viability in cells lacking functional BRCA2.

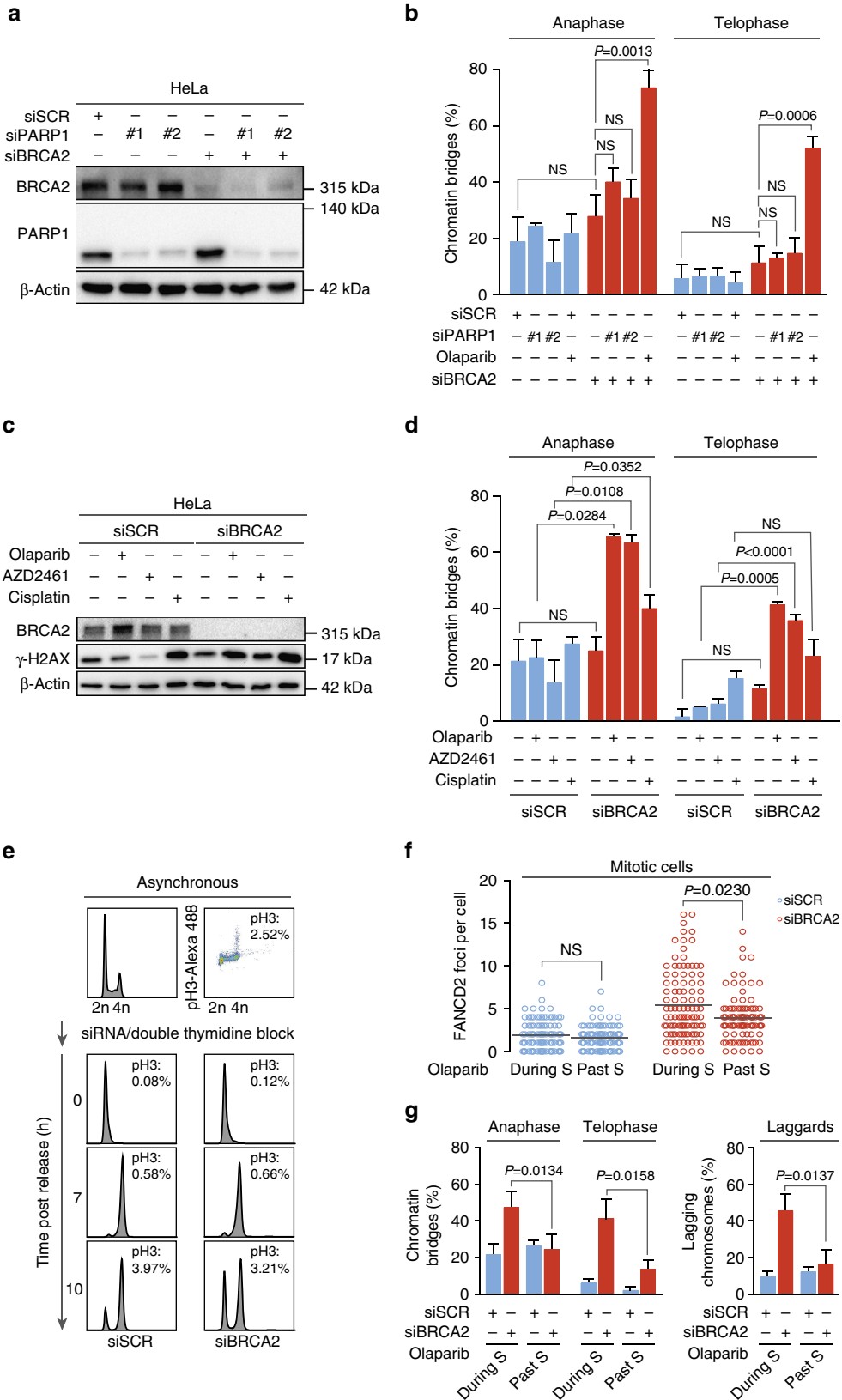

To test whether these observations could be extrapolated *in vivo*, we analysed $Brca2^{-/-};p53^{-/-}$ mammary tumours, generated in $K14cre;Brca2^{F/F};p53^{F/F}$ mice, orthotopically transplanted into syngeneic wild-type (wt) mice, and treated with vehicle or olaparib for 28 days (Fig. 5d). Notably, remnants of olaparib-treated $Brca2^{-/-};p53^{-/-}$ tumours showed significantly increased numbers of multinucleated cells (Fig. 5d,e). To test if this phenotype is generic for HR-deficient tumours, we also analysed $Brca1^{-/-};p53^{-/-}$ mammary tumours, derived from $K14cre;Brca1^{F/F};p53^{F/F}$ mice (Fig. 5f). Again, tumour remnants of olaparib-treated $Brca1^{-/-};p53^{-/-}$ tumours showed significantly increased numbers of multinucleated cells (Fig. 5f,g). Taken together, these data confirm PARP-inhibitor-induced multinucleation in BRCA1- or BRCA2-deficient tumour cells, and suggest that failed mitosis may contribute to the cytotoxicity of PARP inhibition in these cancer cells.

**Mitotic progression promotes PARP-inhibitor cytotoxicity.** We next investigated to what extent the progression through mitosis contributes to the cytotoxic effects of PARP inhibitor treatment in these tumour cells. To address this, we aimed to prevent progression through mitosis, while still allowing for DNA replication. To this end, we inactivated the early mitotic inhibitor-1 (EMI1). During interphase, EMI1 keeps the APC/C E3 ligase inactive, and thereby allows for the accumulation of numerous mitotic regulators, including B-type cyclins[36]. Inactivation of EMI1 leads to premature APC/C activation in G2-phase, interferes with cyclin B accumulation and consequently precludes mitotic entry. As a result, EMI1-depleted cells bypass mitosis and enter cycles of endoreplication[37]. Indeed, following EMI1 depletion, HeLa cells were unable to enter mitosis, yet continued DNA replication as judged by a large population of endoreplicating cells (>4n DNA content) by flow cytometry (Fig. 6a,b). Subsequently, HeLa cells were depleted for BRCA2 either alone or in combination with EMI1, and induction of apoptosis was analysed by annexin-V staining (Fig. 6c). Whereas olaparib treatment resulted in clear induction of apoptosis in BRCA2-depleted cells, co-depletion of EMI1 rescued the induction of apoptosis (Fig. 6c,d). Next, we assessed whether these effects translated into altered short-term cell survival. Interestingly, olaparib sensitivity of BRCA2-depleted cells was largely nullified by concomitant EMI1 depletion (Fig. 6e). These observations were confirmed in $BRCA2^{-/-}$ DLD-1 cells, in which PARP inhibitor sensitivity was rescued when mitosis was bypassed due to EMI1 depletion (Fig. 6f,g). Importantly, EMI1 depletion did not alleviate induction of DNA lesions in response to PARP inhibition in BRCA2-depleted cells, as judged by foci analysis of FANCD2 (Supplementary Fig. 7A) and γ-H2AX (Supplementary Fig. 7B). Also, loss of PARP inhibitor sensitivity could not be attributed to decreased proliferation rates, since EdU incorporation was not impaired after BRCA2 and EMI1 co-depletion, when compared to depletion of BRCA2 alone. (Supplementary Fig. 7C). Taken together, our data show that forced bypass of mitosis results in decreased PARP inhibition-induced cytotoxicity, and indicate that progression through mitosis promotes cell death in BRCA2-deficient cancer cells treated with PARP inhibitor.

## Discussion

Our findings on PARP-inhibitor-induced cytotoxicity in HR-deficient cancer cells extend on recent findings that mitotic processing of DNA lesions is linked to genome stability[24,25,38–40]. Further, our data challenge the dogma that accumulation of DNA DSBs due to combined loss of base excision repair and HR is the main contributor to synthetic lethality. While DSBs do occur, other aberrant replication intermediates also arise during replication in HR-deficient cells treated with PARP inhibitors[23–27]. Furthermore, our data show that these lesions do not immediately lead to cell death, but can be transferred into mitosis, resulting in chromatin bridging and subsequent cytotoxicity. Inactivation of HR components BRCA1/2 or RAD51 (ref. 31), or Fanconi Anemia components[30], has been previously linked to mitotic defects. Loss of either BRCA1/2 or RAD51 was found to increase the percentage of cells with ultra-fine anaphase bridges as well as bulky chromatin bridges during anaphase[31], in agreement with our data. Previously, PARP inhibition was shown to promote mitotic aberrancies and multinucleation[41]. Similar to our data, these reports showed that PARP inhibition alone does not appear to induce severe effects on mitosis in wt cells. Rather, our data indicate that severe mitotic defects only arise when PARP inhibitors are combined with a HR defect.

Chromatin bridges were previously described to frequently arise as a consequence of unresolved replication lesions[26]. Since PARP inhibition in HR-deficient cells also leads to replication fork instability, it is conceivable that unresolved replication intermediates underlie the formation of chromatin bridges in HR-deficient cells upon PARP inhibition. Although the exact nature of these lesions remains obscure, the observed FANCD2-positive foci in mitosis suggest that under-replicated regions may persist after replication fork stalling. Indeed, secondary mutations that rescue replication fork stability in BRCA2-deficient cancer cells, render them resistant to PARP inhibition[22].

Late-stage replication intermediates that persist up until mitosis are normally cleared by DNA resolvases[40,42]. Specifically, the structure-specific endonuclease complex MUS81-EME1 operates in conjunction with SLX4 and GEN1 to resolve DNA joint molecules, and these enzymes are known to be highly active during mitosis and are required for proper chromosome segregation[39,43]. One could therefore speculate

---

**Figure 3 | PARP trapping during S-phase is required for mitotic chromatin bridge formation.** Throughout the figure blue bars represent BRCA1/2 proficient cells, red bars represent BRCA1/2 deficient cells. (**a**) HeLa cells were transfected with indicated siRNAs, and immunoblotting for BRCA2, PARP1 and β-Actin was done at 48 h after transfection. Lines next to blots indicate positions of molecular weight markers. (**b**) HeLa cells were transfected with indicated siRNAs and after 24 h were treated with olaparib (0.5 μM) or DMSO for 24 h. Percentages of cells containing chromatin bridges (n > 20 events per condition) were quantified. *P* values were calculated using two-tailed Student's *t*-test. (**c**) HeLa cells were transfected with indicated siRNAs, and after 24 h were treated with olaparib (0.5 μM), AZD2461 (1 μM), cisplatin (1 μM) or DMSO for another 24 h. Immunoblotting was performed for BRCA2, γ-H2AX and β-Actin (**d**) HeLa cells were treated as for **b**. Percentages of cells containing chromatin bridges (n > 20 events per condition) were quantified. *P* values were calculated using two-tailed Student's *t*-test. (**e–g**) HeLa cells were transfected with indicated siRNAs and after 24 h were synchronized in the G1/S-phase border by a double-thymidine block. Cells were either treated with 0.5 μM olaparib for 3 h directly after release from the thymidine block ('during S' treatment) or 7 h after release ('past S' treatment). Representative flow cytometry images are presented (**e**). Cells were then fixed and assessed for mitotic FANCD2 foci (**f**), chromatin bridges and lagging chromosomes (**g**). *P* values for FANCD2 foci were calculated using the two-tailed Mann–Whitney test. The number of cells containing chromatin bridges (n > 20 events per condition) and lagging chromosomes (n > 40 events per condition) were quantified. *P* values were calculated using two-tailed Student's *t*-test. Throughout the figure 'NS' indicates not significant. All error bars indicate s.d. of three independent experiments.

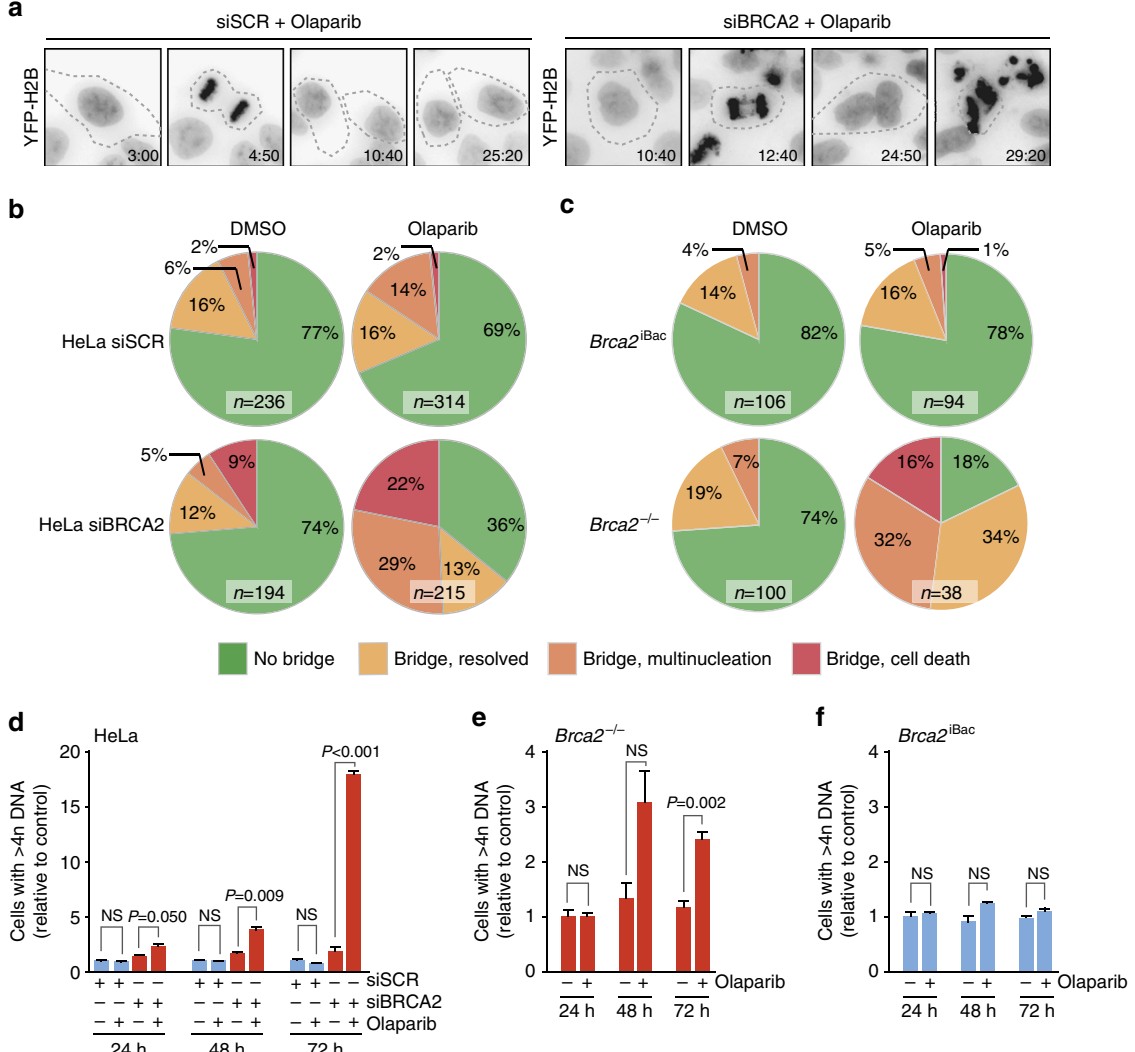

**Figure 4 | Chromatin bridges cause multinucleation and cell death.** (**a**) HeLa cells stably expressing YFP-H2B were transfected with indicated siRNAs for 24 h and subsequently treated with olaparib (0.5 μM) for 24 h. Subsequently, cells were analysed by live-cell microscopy for 36 h. Representative YFP-H2B images are shown. Dotted lines indicate cell boundaries as based on the phase-contrast images. (**b**) HeLa cells stably expressing YFP-H2B were treated as for **a**. All anaphase cells were scored for the presence of anaphase cells and cell fate was analysed. (**c**) KB2P1.21 ($Brca2^{-/-}$) and KB2P1.21R1 ($Brca2^{iBac}$) cells were treated with olaparib (0.5 μM) for 24 h after which they were analysed for at least 48 h by live-cell microscopy. All anaphase cells were scored for chromatin bridges and cell fate was analysed. (**d**) HeLa cells were transfected with control (blue) or BRCA2 siRNAs (red) for 24 h and then treated with olaparib (0.5 μM) for 24, 48 or 72 h. Then, cells were fixed and DNA content was analysed by flow cytometry. Indicated percentages show >4n DNA content. $P$ values were calculated using two-tailed Student's $t$-test. (**e,f**) KB2P1.21 ($Brca2^{-/-}$, red bars, **e**) and KB2P1.21R1 ($Brca2^{iBac}$, blue bars, **f**) cells were treated with olaparib (0.5 μM for 24, 48 or 72 h, after which cells were fixed and DNA content was analysed. Percentages of cells with >4n DNA content are indicated. Averages and s.d. from three technical replicates are indicated. $P$ values were calculated using two-tailed Student's $t$-test. Throughout the figure 'NS' indicates not significant.

that the amount of DNA lesions induced by PARP inhibition in HR-defective cells exceeds the resolvase capacity during mitosis, and leads to the accumulation of toxic DNA lesions.

Interestingly, forced mitotic bypass through EMI1 depletion could largely rescue viability of HR-deficient cells upon PARP inhibition. This implies that progression through mitosis facilitates PARP-induced cytotoxicity, at least in short-term assays. Since *EMI1* is an essential gene *in vivo*[44] and is also required for long-term growth *in vitro*[45–47], we do not consider EMI1 downregulation as a clinically relevant means to achieve long-term PARP inhibitor resistance. Rather, EMI1 served as a tool to bypass mitosis without impairing replication. In line with this notion, RNA sequencing analysis of *Brca2* mutant cancers that were either sensitive or resistant to PARP inhibition did not

provide evidence that EMI1 loss is involved in PARP inhibitor resistance (Supplementary Fig. 7D).

In addition, our results suggest that PARP-inhibitor-induced cytotoxicity requires cycles of both replication and mitosis, and that tumour cells that remain in G1- or G2-phase longer are more resistant to PARP-inhibitor-induced cytotoxicity. Conversely, drugs that promote mitotic entry, such as WEE1 or DDR kinase inhibitors, may potentiate PARP inhibitor treatment. Preliminary evidence indeed shows additive effects of combined inhibition of PARP and WEE1 in BRCA2-deficient cells, which warrants further investigation (Supplementary Fig. 8).

Alternatively, targeting the mitotic spindle using tubulin poisons could be an interesting approach to potentiate PARP inhibitor treatment in HR-deficient cells. Further characterization

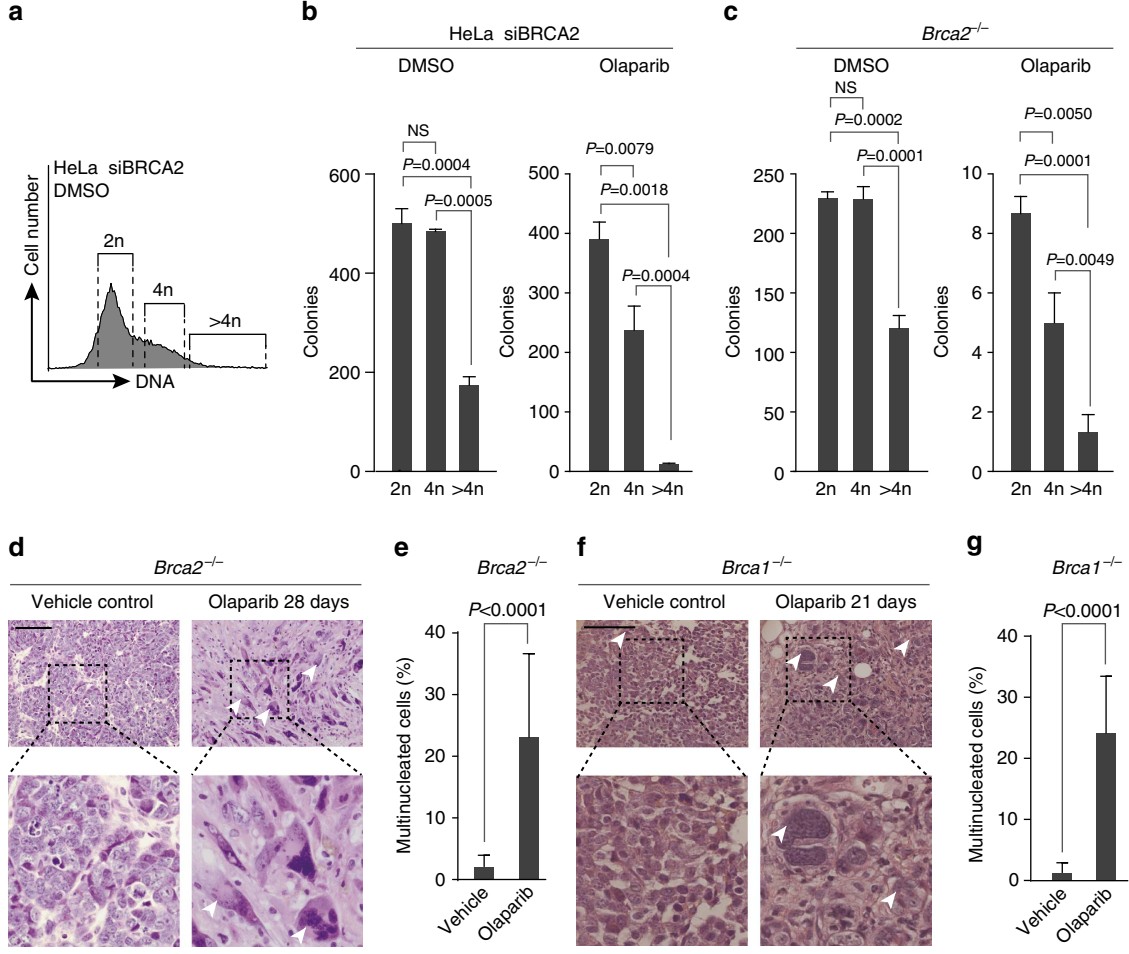

**Figure 5 | Multinucleated BRCA2-deficient cells arising from failed chromatin bridge resolution after PARP inhibitor treatment are not viable.**
(**a**,**b**) HeLa cells were transfected with siRNA targeting BRCA2 for 24 h and were then treated with olaparib (0.5 μM) or DMSO for 72 h. Cells were then incubated with Hoechst for 45 min at 37 °C, after which cells containing 2n, 4n or >4n were sorted as shown in **a**. Subsequently, cells were sorted at a density of 5,000 cells per well in six-wells plates. After 7 days, colony formation was quantified. The graph shows averages and s.d.'s from three replicates. $P$ values were calculated using two-tailed Student's $t$-test. (**c**) KB2P1.21 ($Brca2^{-/-}$) cells were treated, sorted and stained as described for **a** and **b**. After 7 days, colony formation was quantified. The graph shows means with error bars indicating s.d. of three replicates. $P$ values are calculated using two-tailed Student's $t$-test. (**d**) H&E staining of a $Brca2^{-/-};p53^{-/-}$ mammary tumour derived from a tumour-bearing mouse, treated with vehicle or olaparib (50 mg kg$^{-1}$) for 21 days i.p. daily. Arrowheads indicate multinucleated cells. Scale bars represent 100 μm. (**e**) Quantification of the percentage of multinucleated cells in tumours described in **d**. (**f**) H&E staining of a $Brca1^{-/-};p53^{-/-}$ tumour derived from a tumour-bearing mouse, treated with vehicle or olaparib (50 mg kg$^{-1}$) for 21 days i.p. daily. Multinucleated cells and apoptotic cells are indicated. Scale bars represent 100 μm. (**g**) Quantification of the percentage of multinucleated cells in tumours described in **f**. $P$ values were calculated using two-tailed Student's $t$-test. Throughout the figure 'NS' indicates not significant.

of the nature of the DNA lesions that underlie mitotic chromatin bridges and the pathways that respond to these structures is required to elucidate how PARP inhibitor therapy functions at the molecular and cellular level. These insights could then aid in designing rational combination therapies to potentiate PARP inhibitor treatment.

## Methods

**Cell culture and cell cycle synchronization.** HeLa human cervical cancer cells, HEK293T human embryonic kidney cells and human BT-549 and HCC1937 breast cancer cell lines were obtained from ATCC (#CCL2, #CRL3216, #HTB122 and #CRL2336 respectively). DLD-1 human colorectal adenocarcinoma cells were from Horizon (Cambridge, UK). HeLa and DLD-1 cells were cultured in Dulbecco's Modified Eagle's Medium (DMEM). BT-549 and HCC1937 cells were cultured in RPMI medium. Media were supplemented with 10% fetal calf serum (FCS) and cells were cultured at 37 °C in a humidified incubator supplied with 5% $CO_2$. The KB2P1.21 cell line (denoted in the manuscript text as $Brca2^{-/-}$) was established from a mammary tumour from $K14cre;Brca2^{F11/F11};p53^{F2-10/F2-10}$ mice and the KB1P-B11 cell line (denoted in manuscript text as $Brca1^{-/-}$) was established from a mammary tumour from $K14cre;Brca1^{F5-13/F5-13};p53^{F2-10/F2-10}$ mice[7,48].

The KP3.33 cell line (denoted in the manuscript text as $p53^{-/-}$) was established from a mammary tumour from $K14cre;p53^{F2-10/F2-10}$ mice[32]. The KB2P1.21R1 cell line (denoted as $Brca2^{iBAC}$) was generated through stable introduction of an iBAC containing the entire mouse $Brca2$ gene into the KB2P1.21 cell line[32]. All mouse cell lines were cultured in DMEM/F-12 medium, supplemented with 10% FCS, 50 units per ml penicillin, 50 μg ml$^{-1}$ streptomycin, 5 μg ml$^{-1}$ insulin (Sigma), 5 ng ml$^{-1}$ epidermal growth factor (Life Technologies) and 5 ng ml$^{-1}$ cholera toxin (Gentaur), at 37 °C under hypoxic conditions (1% $O_2$, 5% $CO_2$). Cell cycle synchronization was achieved using a double-thymidine block. Specifically, cells were treated with thymidine (2 mM) for 17 h, washed twice with PBS and were incubated in warm medium for 9 h. Subsequently, cells were again incubated in thymidine for 17 h, washed with PBS and released in pre-warmed medium and collected at indicated time points. For treatment of cells during S-phase, cells were treated immediately following release from thymidine. For treatment after S-phase, cells were treated at 7 h after release from thymidine.

**Virus infection.** VSV-G pseudotyped retroviral particles were produced as described previously[49]. In short, HEK293T cells were transfected with 10 μg of indicated pRetroX of pLKO vector, combined with 2.5 μg pMD/p and 7.5 μg pMDg plasmids, expressing the gag/pol and envelop proteins, respectively. The supernatant containing retrovirus was harvested at 48–72 h following transfection,

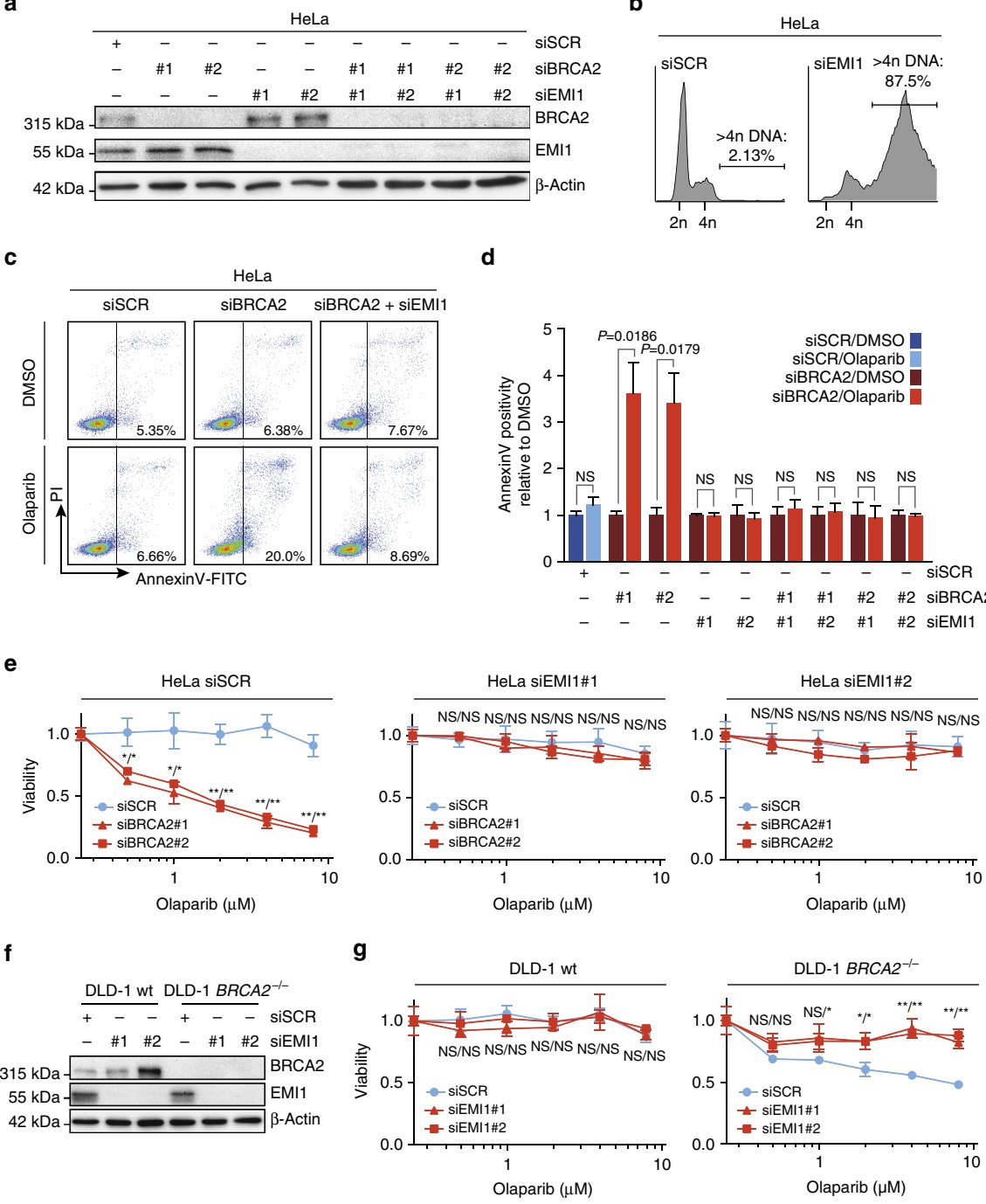

**Figure 6 | Mitotic progression promotes PARP-inhibitor cytotoxicity.** (**a**) HeLa cells were transfected with indicated siRNAs, and immunoblotting for BRCA2, EMI1 and β-Actin was performed at 48 h after transfection. Lines next to blots indicate positions of molecular weight markers. (**b**) HeLa cells were transfected with indicated siRNAs for 48 h, and subsequently fixed. DNA content was analysed by flow cytometry. The percentage of cells containing > 4n DNA is indicated. (**c**) HeLa cells were transfected with BRCA2 siRNAs or control siRNA (SCR), either alone or in combination with EMI1 siRNAs. After 24 h, cells were replated and treated with olaparib (0.5 μM) for 72 h, after which cells were stained with annexin-V-FITC and propidium iodide, and were analysed by flow cytometry. (**d**) HeLa cells were treated and analysed as for **c**. Averages and s.d. of three independent experiments are shown. *P* values were calculated using two-tailed Student's *t*-test. (**e**) HeLa cells were transfected with two independent BRCA2 siRNAs or control siRNA (SCR), either alone or in combination with siRNAs targeting EMI1. After 24 h, cells were replated and allowed to attach for 3 h. Subsequently, cells were treated with indicated concentrations of olaparib for 72 h, and viability was assessed by MTT conversion. Graphs are representative of three independent experiments, and error bars indicate s.d. of three technical replicates. *P* values were calculated using two-tailed Student's *t*-test. (**f,g**) DLD-1 wt or DLD-1 *BRCA2*$^{-/-}$ cells were transfected with two independent siRNAs targeting EMI1 or control siRNA (SCR). After 24 h, cells were replated and incubated for 3 h. Subsequently, cell lysates were immunoblotted for BRCA2, EMI and β-Actin (**f**). In parallel, cells were treated with indicated concentrations of olaparib for 72 h, and viability was assessed by MTT conversion (**g**). Graphs are representative of three independent experiments, and error bars indicate s.d. of three technical replicates. *P* values were calculated using two-tailed Student's *t*-test. Throughout the figure 'NS' indicates not significant, * indicates *P* < 0.05 and ** indicates *P* < 0.01.

was filtered through a 0.45-μM syringe filter and was subsequently used to infect target cells.

Establishment of HeLa cells stably expressing YFP-H2B cells was described previously[27]. In short, HeLa cells were retrovirally infected with pBabe-H2B-YFP, and selected with blasticidine (5 μgram ml[−1], Sigma). To establish KB2P1.21 and KB2P1.21R1 cell lines expressing H2B-EGFP and α-tubulin-mCherry, cells were first transduced with pRetrox-rTTa virus and selected with Geneticin (400 μg ml[−1]). Subsequently, cells were infected with a pRetrox-Tight-Pur virus harbouring H2B-EGFP–T2A-α-tubulin-mCherry and selected in puromycin (1 μg ml[−1]). H2B-EGFP and α-tubulin-mCherry expression was induced by incubation with doxycycline (0.5 μg ml[−1], Sigma).

**MTT assays.** HeLa, DLD-1, BT-549, HCC1937, KB2P1.21 and KB2P1.21R1 tumour cell lines were plated in 96-wells plates. BT-549 cells were pre-treated with 1 μg ml[−1] doxycycline for 48 h. HeLa were plated at 2,000 cells per well, DLD-1 cells at 5,000 cells per well, BT-549 and HCC1937 at 1,000 cells per well, and KB2P1.21 and KB2P1.21R1 were plated at 1,200 cells per well. Cells were allowed to attach for 3 or 24 h and were treated with indicated concentrations of olaparib, MK-1775, or MK4827 (all from Axon Medchem, Groningen, the Netherlands) for 3 or 4 days. Methyl-thiazol tetrazolium (MTT) was added to cells at a concentration of 5 mg ml[−1] for 4 h, after which culture medium was removed and formazan crystals were dissolved in DMSO. Absorbance values were determined using a Bio-Rad benchmark III Biorad microtiter spectrophotometer at a wavelength of 520 nm. Proliferation was determined as the relative decrease in signal compared to DMSO-treated cells. Unless mentioned otherwise, statistical significance was tested using Student's *t*-test.

**RNA interference.** Cells were transfected with 40 nM siRNAs (Ambion Stealth RNAi, Thermofisher) targeting BRCA2 (sequence 1: #HSS186121 and sequence 2: sequence #HSS101095), BRCA1 (sequence 1: #HSS101089 and sequence 2: #HSS186096), RAD51 (sequence HSS1299001), PARP1 (sequence 1: #HSS100243 and sequence 2: #HSS100244), PAXIP1 (encoding PTIP) (sequence #HSS117971), EMI1 (sequence 1: #HSS119992 and sequence 2: #HSS119993) or a scrambled (SCR) control sequence (sequence #12935300) with oligofectamine (Invitrogen) by using manufacturer's guidelines. Alternatively, cells were transduced with Tet-pLKO-puro vectors, for doxycycline-inducible expression of shRNAs (Addgene plasmid #219125, a kind gift from Dmitri Wiederschain[50]. The shRNA sequences are: Luciferase ('LUC'): 5′-AGAGCTGTTTCTGAGGAGCC-3′, BRCA1: 5′- CCC TAAGTTTACTTCTCTAAA-3′ and BRCA2: 5′-AACAACAATTACGAACCAA ACTT-3′. shRNAs were induced using 1 μg ml[−1] doxycycline (Sigma) for 48 h.

**Western blotting.** Cells were lysed using mammalian protein extraction reagent (Thermo Scientific), supplemented with protease inhibitor and phosphatase inhibitor cocktail (Thermo Scientific). Protein content was determined with a Bradford assay after which 20 μg of protein sample was separated by sodium dodecyl sulfate (SDS)/PAGE, transferred to polyvinylidene fluoride (immobilon) membranes and blocked in 5% skimmed milk (Sigma) in Tris-buffered saline (TBS) containing 0.05% Tween20 (Sigma).

Immunodetection was performed with antibodies directed against BRCA2 (Calbiochem, #OP95), BRCA1 (Cell Signaling, #9010), RAD51 (GeneTex, #gtx70230), EMI1 (Invitrogen, #37-6600), γ-H2AX (Cell Signaling, #9718), PTIP (Abcam, ab70434) all diluted 1:1,000 and Beta-Actin (MP Biomedicals, #69100) diluted 1:10,000. Horseradish peroxidase-conjugated secondary antibodies (DAKO) were diluted 1:2,500 and used for visualization using chemiluminescence (Lumi-Light, Roche Diagnostics) on a Bio-Rad bioluminescence device, equipped with Quantity One/ChemiDoc XRS software (Bio-Rad). Uncropped versions of all western blots can be found in Supplementary Fig. 9.

**Flow cytometry.** For apoptosis analysis by annexin V staining, total cell populations were collected by trypsinization and stained with annexin-V-FITC (1:20) and propidium iodide as per manufacturer's instructions (Immune Quality Products). Cells were then analysed on a LSR-II (Becton Dickinson) cytometer using FACSDiva software (Becton Dickinson). For cell cycle analysis, BrdU and phospho-HistoneH3 analysis, cells were fixed in ice-cold 70% ethanol or methanol for at least 6 h and were then immunostained with an Alexa-488-conjugated antibody targeting BrdU (MoBU1, #B35130, 1:200), or anti-phospho-histone-H3 (Ser10, Cell Signaling, #9701, 1:300) in combination with Alexa-488-conjugated secondary antibodies (1:300). DNA staining was performed using propidium iodide in the presence of RNAse. At least 10,000 events per sample were analysed on a FACScalibur (Becton Dickinson). Data was analysed using FlowJo software.

**Immunofluorescence microscopy.** HeLa, KB2P1.21 and KB2P21R1 cells were seeded on glass coverslips in six-well plates. When indicated, HeLa cells were transfected with siRNAs for 48 h or were treated with olaparib (0.5 μM) for 24 h as indicated. If indicated, KB2P1.21 and KB2P21R1 cells were irradiated (5 Gy) using a CIS International/IBL 637 caesium[137] source (dose rate: 0.010124 Gy s[−1]). Cells were fixed using 4% formaldehyde or paraformaldehyde in PBS, and subsequently permeabilized for 5 min in PBS with 0.1% Triton X-100. After extensive washing,

cells were stained with antibodies targeting α-Tubulin (Cell Signaling, #2125, 1:100), RAD51 (GeneTex, #gtx70230, 1:400), FANCD2 (SantaCruz Biotechnology, #sc20022, 1:200) or γ-H2AX (Cell Signaling, #9718, 1:200), in combination with Alexa-488 or Alexa-647-conjugated secondary antibodies (1:300), and were counterstained with DAPI. Early anaphases in which chromosome packs were separated less than 10 nm were excluded for analysis. Anaphase and telophase cells were distinguished based on α-Tubulin staining. Images were acquired on a Leica DM6000B microscope using a × 63 immersion objective (PL S-APO, numerical aperture: 1.30) with LAS-AF software (Leica).

**Live-cell microscopy.** KB2P2.21 and KB2P2.21R1 transduced with H2B-EGFP-IRES-α-tubulin-mCherry and HeLa cells transduced with H2B-EGFP were seeded in eight-chambered cover glass plates (Lab-Tek-II, Nunc) at 10,000 cells per well. Cells were then treated with 0.5 μM olaparib at 24 h before imaging, and were followed for at least 36 h on a DeltaVision Elite microscope, equipped with a CoolSNAP HQ2 camera and a × 40 immersion objective (U-APO 340, numerical aperture: 1.35). Images were obtained each 10 min, with 12 images being acquired in the Z-axis, at 0.5 μm interval. Image analysis was done using SoftWorX software (Applied Precision/GE Healthcare). The fate of all cells that entered mitosis and proceeded at least until anaphase were included for analysis.

**DNA fibre analysis.** To assess replication fork protection. HeLa cells were pulse-labelled with CIdU (25 μM) for 60 min. Next, cells were washed with medium and incubated with hydroxyurea (HU, 5 mM) for 5 h. Cells were harvested using trypsine and lysed on microscopy slides in lysis buffer (0.5% SDS, 200 mM Tris (pH 7.4), 50 mM EDTA). DNA fibres were spread by tilting the slide and were subsequently air dried and fixed in methanol/acetic acid (3:1) for 10 min. For immunolabelling, spreads were treated with 2.5 M HCl for 1.5 h. CIdU was detected by staining with rat anti-BrdU (1:1,000, AbD Serotec) for 1 h and was further incubated with AlexaFluor 488-conjugated anti-rat IgG (1:500) for 1.5 h. Images were acquired on a Leica DM-6000RXA fluorescence microscope, equipped with Leica Application Suite software. The lengths of CIdU and IdU tracks were measured blindly using ImageJ software. Statistical analysis was performed using two-sided Mann–Whitney tests with 95% confidence intervals.

**Generation of mammary tumours.** Brca1[−/−];p53[−/−] and Brca2[−/−];p53[−/−] mammary tumours were generated in K14cre;Brca1[F/F];p53[F/F] and K14cre;Brca2[F/F]; p53[F/F] mice, respectively, genotyped, and orthotopically transplanted into syngeneic wt mice as described[7]. Starting 2 weeks after tumour grafting, in female FVB/N mice (6–8 weeks old), the onset of tumour growth was checked at least three times per week. Mammary tumour size was determined by caliper measurements. When mammary tumours reached a size of ∼200 mm[3], treatment was initiated. Olaparib was used by diluting 50 mg per ml stocks in DMSO with 10% 2-hydroxyl-propyl-β-cyclodextrine/PBS such that the final volume administered by intraperitoneally (i.p.) injection was 10 μl g[−1] of body weight. Olaparib (50 mg kg[−1]) was given i.p. daily for 21 or 28 consecutive days. Controls were dosed with vehicle only. Animals were killed with CO₂ at the end of treatment when the minimal residual disease stage was reached. At this point, olaparib-treated tumour explants had an approximate size of 1 mm[3], whereas control-treated tumour explants had a volume of ∼1 cm[3]. Tumour samples were fixed in 4% formaline and processed for hematoxylin/eosin staining. S.d. represent 10 different fields, containing at least 100 cells. All experimental procedures on animals were approved by the Animal Ethics Committee of the Netherlands Cancer Institute.

**RNA sequencing and analysis.** Fresh-frozen tumour tissues of AZD2461-sensitive (n = 23) and AZD2461-resistant (n = 36) Brca2[−/−];p53[−/−] tumours (described in (ref. 22)), were placed in 1 ml of TRIsure reagent (Bioline) and subjected to mechanical disruption with Tissue Lyser LT (Qiagen, oscillation: 50 s[−1], time: 10 min). Homogenized lysates were further processed for RNA isolation following TRIsure manufacturer's protocol. Quality and quantity of the total RNA was assessed by the 2100 Bioanalyzer using a Nano chip (Agilent, Santa Clara, CA). Total RNA samples having RIN > 8 were subjected to library generation.

Strand-specific libraries were generated using the TruSeq Stranded mRNA sample preparation kit (Illumina Inc., San Diego, RS-122-2101/2), according to the manufacturer's instructions (Illumina, Part #15031047 Rev. E). Briefly, polyadenylated RNA from intact total RNA was purified using oligo-dT beads. Following purification, the RNA was fragmented, random primed and reverse transcribed using SuperScript II Reverse Transcriptase (Invitrogen, part #18064-014) with the addition of Actinomycin D. Second strand synthesis was performed using Polymerase I and RNaseH with replacement of dTTP for dUTP. The generated cDNA fragments were 3′-end adenylated and ligated to Illumina Paired-end sequencing adapters and subsequently amplified by 12 cycles of polymerase chain reaction. The libraries were analysed on a 2100 Bioanalyzer using a 7500 chip (Agilent, Santa Clara, CA), diluted and pooled equimolar into a 10 nM sequencing stock solution. Illumina TruSeq mRNA libraries were sequenced with 50 base single reads on a HiSeq2000 using V3 chemistry (Illumina Inc., San Diego).

The resulting reads were trimmed using Cutadapt (version 1.12)[8] to remove any remaining adapter sequences, filtering reads shorter than 20 bp after trimming to ensure efficient mapping. The trimmed reads were aligned to the GRCm38 reference genome using STAR (version 2.5.2b)[9]. QC statistics from Fastqc (version 0.11.5) and the above-mentioned tools were collected and summarized using Multiqc (version 0.8)[51]. Gene expression counts were generated by featureCounts (version 1.5.0-post3)[52], using gene definitions from Ensembl GRCm38 version 76. Normalized expression values were obtained by correcting for differences in sequencing depth between samples using DESeq median-of-ratios approach[53], and subsequent log-transformation the normalized counts.

**Data availability.** The data that support the findings of this study are available from the corresponding author upon reasonable request. RNA sequence data has been deposited at the European Nucleotide Archive (ENA) under accession number PRJEB20535.

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

## Acknowledgements

We are grateful to van Vugt lab members for constructive comments. We thank H.R. de Boer for technical assistance and critical review of the manuscript, G. Berger for technical assistance and A. Bhattacharya for help with statistics. Financial support came from the Dutch Cancer Society (RUG 2011–5093 to M.A.T.M.v.V. and NKI 2011–5220 to J.J. and S.R.), the Netherlands Organization for Scientific Research (NWO-VIDI 916–76062 to

M.A.T.M.v.V.), the European Research Council (ERC-CoG- 682421 to M.A.T.M.v.V. and ERC-CoG-681572 to S.R.), the Swiss National Science Foundation (310030_156869 to S.R.) and Swiss Cancer Research (MD-PhD fellowship MD-PhD-3446-01-2014 to S.B.).

## Author contributions

P.M.S. and M.A.T.M.v.V. conceived the project and analysed data. P.M.S., F.T. and C.S. performed *in vitro* experiments. F.F. assisted with live-cell microscopy and establishing reporter cell lines. E.G. and S.R. performed *in vivo* experiments and analysed RNAseq data. P.M.S., F.T., C.S., E.G., S.B., S.R. and M.A.T.M.v.V. analysed data. A.M.H., P.B., J.J. and M.T. established or provided reagents. P.M.S. and M.A.T.M.v.V. wrote the manuscript. All authors assisted in editing the manuscript and approved it before submission.

## Additional information

**Competing interests:** The authors declare no competing financial interests.

**Publisher's note**: 

