## [Peer Review File · Nature Communications]

Reviewers' comments:

Reviewer #1 (Remarks to the Author):

PARP1 is a major enzyme that catalyses poly(ADP-ribose) polymer formation in response to DNA single strand breaks (SSBs) and is a component of the base excision repair (BER) machinery. PARP1 has also a role in the stabilisation of replication forks in S-phase cells. The synthetic lethality concept has been developed based on the function of PARP1 in the SSBs and replication forks, a failure of which induces cytotoxic double strand breaks (DSBs). Thus, PARP inhibitors (Pi) have been developed to kill cancer cells, which are deficient in the homologous recombination repair (HRR) of DSBs, for example, in BRCA1/2 deficient cancers.

In this manuscript, Schoonen et al. attempted to study the specific mechanism underlying the synthetic lethality of Pi and BRCA mutant cancer cells. The authors found that Pi induces specifically mitotic abnormalities (chromatid bridges, lagging chromosomes as well as multinucleation) in BRCA-deficient Hela cells as well as in mouse mammary cancer cell lines. This study thus provides further evidence and supports the idea of how Pi kills HRR deficient cancer cells, namely by chromosome catastrophe. Overall, these findings are useful for understanding the molecular causes of the PARP inhibition-induced cell death of BRCA-deficient cells. However, the mechanism of the synthetic lethality in this context has been well studied and the current study presents only limited advance.

Major comments

1. The authors proposed that the chromosome catastrophe is the mechanism of cell death by Pi-induced synthetic lethality. The authors concluded that these indeed originated from DSBs which are carried on to mitosis. The original cause of these DSBs is indeed from replications, which have been demonstrated by others and well accepted in the field.
2. Many of the data represents two sets of cell lines, namely Hela cells after the inactivation of BRCA and mouse BRCA2^{-/-} mammary carcinoma cell lines. These data confirmed each other for their findings. The most used cells in this aspect may include human breast cancer cells, such as HCC1937.
3. Fig 1C,D,F,G. The color code of the bars is missing.
4. Fig 3C and 3D-G are redundant to Fig 1B and Fig 2. One set of the data can be moved to Supplementary information.
5. Fig 4B,C are a summary of the data. The arrows are unclear and misleading. It would be useful to show the original data.
6. Fig 5. Can the authors verify that the low colon numbers in (>4N) the cells are due to cell death and not due to proliferation arrest? It is well known that multinucleated cells do not necessarily die if they do not try to divide.
7. Fig 5D,F. The size of the tumour explants should be shown. Is there an increase of apoptotic cells in histologies?
8. Statistical analyses are missing in Fig 1C,D,G,G; Fig 2D-G, I, J; Fig S1B,C; S4C.

Reviewer #2 (Remarks to the Author):

In this paper the authors are showing that PARP1 inhibition in BRCA2 deficient cells leads to problems during the next mitosis (FANCD2 foci, anaphase bridges, multi-nucleated cells). Importantly the results are confirmed with cell lines derived from patients and mouse models. It should be noted that the formation of mitotic problems in absence of BRCA1/2 and RAD51 is not new and has been already reported as indicated by the authors (Laulier et al. 2011 NAR). In contrast the fact that PARP1 inhibition in the context of BRCA2 impairment strongly increases the problems during the next mitosis is new. This result is really convincing and is demonstrated using either siRNA directed against BRCA2, BRCA2^{-/-} cell lines from patients complemented with a BRCA2 BAC and BRCA2^{-/-} cell lines obtained using CRISPR-Cas9 with different methods. They also demonstrate that the synthetic lethality (SL) between PARP1 inhibition and BRCA2 is due to

problems during mitosis, this is done using a method to bypass mitosis (EMI1 depletion). With this approach they are able to suppress synthetic lethality between BRCA2 and PARP1 which is obviously interesting in term of resistance to PARP1 inhibitors in patients. Overall this is a nice study which deserves to be published in Nature Communications. However I have few questions and comments which need to be adressed before publication.

Major comments

It has been shown that PARP1 inhibitors lead to PARP1 trapping on the DNA (Murai et al. 2012 Cancer Research) that is likely induces SL. Therefore the situation could be different with siRNA against PARP1. I wonder if the authors tried to co-deplete PARP1 and BRCA2 and check if the SL is still due to problems during mitosis?

EMI1 inactivation suppresses the SL between BRCA2 and PARP1. Did they look if EMI1 level is affected in BRCA2 tumors resistant to PARP inhibitors?

As they suggested in the end of the discussion WEE1 inhibitors could potentiate the effect of PARP inhibitors in BRCA2 mutant cells. Did they try to test this? It would reinforce the role of mitosis.

Minor Comments

I have several issued about data presentation and statistical analysis which need to be adressed.

Figure 1E-F: The legend does not explain what the value between brackets is, I assume it is median or average but it should be explained.

Figure 2: instead of showing representative experiments they could show all the repetitions with standard deviation on the graph. It would be more convincing even though the results are really clear.

Figure 3: H, I and J legends are missing, again it might be better to show the repetitions on the experiments with standard deviation.

Figure 4: In the text it is indicated Figures 4E-I and there is no I panel, the authors needs to correct this. In Figure 4G there is no statistical difference at 48hrs but there is now at 72hrs even though the differences are similar, there must be a problem here. I assume the standard deviations represent repetitions but is not written in the legend.

References 28 and 35 are the same

Reviewer #3 (Remarks to the Author):

In this manuscript, Schoonen et al. investigate the mechanisms of PARP inhibitor induced cytotoxicity in HR-deficient cancer cells, with a particular focus on the role of replication fork and mitotic alterations. The authors show that PARP inhibition leads to reduced fork protection and increased FANCD2 foci in the setting of BRCA2 deficiency. The authors then focus on the mitotic consequences of PARP inhibition, demonstrating that PARP inhibition exacerbates chromatin bridge and lagging chromosome formation in human cancer cells depleted for BRCA2, BRCA1, or RAD51. Using live-cell imaging, these chromatin bridges can be seen persisting and leading to polyploidy and death. Similar phenotypes are seen across multiple mouse and human lines with BRCA2 knockout and knockdown. In vivo, PARP inhibitor treated Brca2^{-/-} and Brca1^{-/-} mammary tumors also displayed increased polyploidy. Finally, the authors provide evidence that progression through

mitosis, which they modulate by EMI1 depletion, is necessary for PARP inhibitor cytotoxicity.

This study helps clarify the mechanisms of PARP inhibitor induced cytotoxicity and provides findings of potential clinical relevance to PARP inhibitor sensitivity and resistance in HR-deficient cancers. A few clarifications would strengthen the specificity and mechanistic underpinnings of this work. This is important considering that many of the findings in this manuscript are somewhat expected.

Comments:

1I There are existing papers in literature (not cited by the authors) reporting (i) that DNA replication stress-associated lesions are transmitted to S-phase, resulting in chromosomal mis-segregation and failed cytokinesis (Ichijima et al., Plos One, 2010) (ii) Cells deficient in Fanconi Anemia proteins (hence enhanced replication stress) display an increased number of inter-chromatid bridges resulting in failed cytokinesis and subsequent cell death via apoptosis (Vinciguerra et al, 2010, J Clin Invest) (iii) Loss of ZNF365, a protein involved in resolving replication stress at fragile sites leads to formation of anaphase bridges and cytokinesis failure (Zhang et al., 2013, Cancer Discovery). Based on the previous study (Vinciguerra et al., 2010, J Clin Invest), the claim made by the authors that "evidence that impaired replication directly results in cell death is lacking" is not true. Overall, it is already known that increasing replication stress, which in this manuscript was done by treating HR-deficient cells with PARPi, leads to the mis-segregation of chromosomes, failed cytokinesis and subsequent cell death.

2) In Fig 1-3, olaparib is the only inhibitor used. Would similar results be expected with other PARP inhibitors and platinum agents? Is this unique to PARP inhibition or a general result of damage accumulation?

3) The authors suggest that aberrant replication intermediates caused by PARP inhibition persist into mitosis and that these intermediates are the cause of chromatin bridge formation and cytotoxicity. While this is plausible, PARP inhibition during mitosis could also directly affect the repair of DNA lesions encountered during this phase. To address these possibilities, cell cycle synchronizations with shorter PARP inhibitor treatments could be used.

4) If the compromised fork stability leads to mitotic DNA damage and hence failed cytokinesis, then it is important to demonstrate that this mitotic DNA damage generated by PARPi can be rescued by protecting the replication fork by either inhibiting Mre-11 or knocking out Ptip1.

5) What is the fate of FANCD2+ cells resulting from PARP inhibitor treatment? Do the ones with high numbers of foci have a propensity to become multinucleated or die?

6) Would the authors expect an increase in DSBs/gH2AX foci in interphase and mitotic cells after PARP inhibitor treatment? How would this compare to FANCD2 foci?

7) Did cell death from olaparib in multinucleated cells occur during the subsequent interphase or during a subsequent mitosis? This type of cell death was frequent in HR-deficient cells treated with olaparib. If the multinucleation and accumulation of replication intermediates were directly detrimental to cells, one would expect EMI1 depleted cells to be sensitive to PARP inhibition. However, the lack of sensitivity may point to a direct role of PARP during mitotic repair. This could be addressed with experiments outlined in Comment 2.

8) Does PARP inhibition still lead to DNA damage (FANCD2 foci) in HR-deficient cells depleted for EMI1? On a similar note, can the authors use a method of mitotic inhibition that does not cause multinucleation? While this is a good control for allowing continued replication, the mechanism of multinucleation itself may confer some degree of resistance.

Minor points:

- (i) The nomenclature SCR for "siRNA targeting control sequence" is confusing. It will be helpful to just use the commonly used term "control siRNA".
- (ii) Olaparib is misspelled in Fig 1E and Fig 1F
- (iii) Fig2a: The DAPI signal is saturated and the tubulin signal is diffused in the cytoplasm rather than being concentrated in the spindle.
- (iv) Fig 3a-3c: The results presented are already known in the literature and hence they can be moved to the Supplementary section.
- (v) Figure 4e: 1.35% rather than 1,35%
- (vi) Abbreviation:DSBs needs to be defined in the first paragraph rather than in the second paragraph.
- (vii) Pg9. First paragraph: Define "wt" in text or use "wild-type"

Point-by-point rebuttal

Reviewer #1

We thank the reviewer for the constructive comments.

Comment 1.

"The authors proposed that the chromosome catastrophe is the mechanism of cell death by Pi-induced synthetic lethality. The authors concluded that these indeed originated from DSBs which are carried on to mitosis. The original cause of these DSBs is indeed from replications, which have been demonstrated by others and well accepted in the field."

Reply:

We indeed propose that mitotic catastrophe is involved in PARP inhibitor-induced cytotoxicity in HR-deficient cancer cells. However, we actually do not suggest that this phenotype originates from replication-associated DSBs. Although we show that the DNA lesions originate in S-phase (and actually require S-phase progression), we think that unresolved stalled replication forks (as indicated by FANCD2-positive foci) lie at the basis of the mitotic failures. We have added novel data showing that PARP inhibition causes only minor increases in γ -H2AX foci, in contrast to other genotoxic agents such as cisplatin (new Supplemental Figures 1C, D and new Figures 3C, D). Yet, the numbers of FANCD2 foci during interphase increase much more robustly, and these lesions remain unresolved even until mitosis. Also, we have included novel data shown that those mitoses with FANCD2 foci actually result in mitotic aberrancies (new Figures 3E, F, G and supplemental Figure 4C).

We agree with the reviewer that PARP inhibitors have indeed previously been acknowledged to interfere with DNA replication (as was already referenced in the original manuscript). However, we feel that the observation that PARP inhibitor treatment causes DNA lesions that persist into mitosis is new, and the observation that this feature promotes cell death has not been reported to date.

Comment 2.

"Many of the data represents two sets of cell lines, namely HeLa cells after the inactivation of BRCA1 and mouse BRCA2-/- mammary carcinoma cell lines. These data confirmed each other for their findings. The most used cells in this aspect may include human breast cancer cells, such as HCC1937."

Reply:

We have used three models in our manuscript, that all gave very similar results. Firstly, we used tumor cells derived from genetically engineered mouse models with a defined defect in *Tp53* and *Brca1* or *Brca2*. Secondly, we used the human HeLa cervical cancer cell line in which we used transient inactivation of BRCA1/2 using siRNA. Thirdly, we used the human colon cell line DLD1, in which CRISPR/Cas9 was used to induce a permanent BRCA2 mutation.

As suggested, we have now included experiments with the human breast cancer cell line HCC1937, which harbors a BRCA1 null mutation as well as the hypomorphic BRCA1 5382insC allele. Analysis of HCC1937 cells showed very similar levels of mitotic aberrancies upon PARP inhibitor treatment (new Figures 2D and Supplemental Figure 2H). Since the HCC1937 cell line harbors a hypomorphic BRCA1 allele, we also validated the observed effects in breast cancer cell lines, in which we depleted BRCA1 or BRCA2. To this end, we engineered the human HR-proficient breast cancer cell line BT-549 with doxycycline-inducible shRNAs targeting BRCA1 or BRCA2 (new Figures 2C and Supplemental Figure 2E, F and G). Upon treatment with PARP inhibitor, and concomitant addition of doxycycline in shBRCA1/2 cells, but not in control cell lines, we again observed robustly increased levels of mitotic aberrancies. These results show that our observation also remain valid in human breast cancer cell lines, either harboring cancer-associated BRCA1 mutations, or shRNA-mediated BRCA1/2 inactivation.

Comment 3. *"Fig 1C,D,F,G. The color code of the bars is missing."*

Reply: We apologize for the color code being absent. We have adjusted the figure and legends.

Comment 4. *"Fig 3C and 3D-G are redundant to Fig 1B and Fig 2. One set of the data can be moved to Supplementary information."*

Reply: We have rearranged the figures and have moved the suggested panels to Supplemental Figures.

Comment 5. *"Fig 4B,C are a summary of the data. The arrows are unclear and misleading. It would be useful to show the original data."*

Reply: We regret that the way of plotting the data is perceived as misleading. We have now displayed the data differently in a new panel in Figure 4B, and have shown original data in the new Supplemental Figure 5A.

Comment 6. *"Fig 5. Can the authors verify that the low clone numbers in (>4N) the cells are due to cell death and not due to proliferation arrest? It is well known that multinucleated cells do not necessarily die if they do not try to divide."*

Reply: We have repeated the assays from Figure 5 and have included BrdU analysis to assess replication rates (now included in Suppl. Figure 6D and 6E). These data show that replication rates do not drop in cultures with higher levels of multinucleated cells (Suppl. Figure 6D). Within these samples, we next analyzed replication in cells with 'normal' ploidy versus cells with increased ploidy. When BrdU positivity of S-phase cells with 'normal' ploidy (DNA content between 2n and 4n) was compared with cells having DNA content between 4n and 8n, we observed a statistically significant but minor drop (Suppl. Figure 6E). These data indicate that multinucleated cells may have somewhat hampered replication, but do not show a dramatic drop in replication that would account for the observed loss of clonogenic outgrowth.

Comment 7. *"Fig 5D,F. The size of the tumour explants should be shown. Is there an increase of apoptotic cells in histologies?"*

Reply: the size of the tumor explants was approximately 1mm³ in case of olaparib-treated animals, and 10x10x10 mm³ in case of control-treated animals. These experimental details have now been added to the Supplemental Document, where we also refer to the publication by Rottenberg *et al*, (PNAS, 2008), in which the experimental design has been extensively described, and the increased amounts of apoptotic cells have been analyzed.

Comment 8. *“Statistical analyses are missing in Fig 1C,D,G,G; Fig 2D-G, I, J; Fig S1B,C; S4C.”*

Reply: We have now included statistical analyses of the indicated figures.

Reviewer #2

We thank the reviewer for the positive feedback and constructive comments.

Comment 1: *“It has been shown that PARP1 inhibitors lead to PARP1 trapping on the DNA (Murai et al. 2012 Cancer Research) that is likely induces SL. Therefore the situation could be different with siRNA against PARP1. I wonder if the authors tried to co-deplete PARP1 and BRCA2 and check if the SL is still due to problems during mitosis?”*

Reply: This is an important issue, and it has indeed been previously suggested that trapping activity of PARP inhibitors underlies the synthetic lethality in HR-deficient cancers. To address this point, we have repeated the experiments with two independent PARP1 siRNAs (new Figures 3A, 3B). We were able to efficiently deplete PARP1 levels, but this did not lead to a phenocopy of the results obtained with olaparib or with PARP inhibitor AZD2461 (now included in new Figures 3C, 3D). These data underscore that PARP trapping is required to induce mitotic defects, and indeed suggests that the SL between PARP inhibitors and HR deficiency is due to defective mitosis.

Comment 2: *“EMI1 inactivation suppresses the SL between BRCA2 and PARP1. Did they look if EMI1 level is affected in BRCA2 tumors resistant to PARP inhibitors?”*

Reply: Emi1 is an essential gene, as recently shown in multiple high-throughput screens (Blomen *et al*, Science 2015; Wang *et al*, Science 2015; Hart *et al*, Cell 2015). As we already put forward in the discussion section of the manuscript, we have used Emi1 inactivation merely as a tool to bypass mitosis. Although cells can be transiently depleted of Emi1 to study cells that do not undergo mitosis, Emi1 depletion precludes clonogenic survival of cells. Therefore, we previously stated that we did not expect Emi1 inactivation to be a likely resistance mechanism. To test this, we have now analyzed RNA-seq data of *Brca2*^{-/-}; *Tp53*^{-/-} mouse mammary tumors that were PARP inhibitor-sensitive versus tumor with acquired PARP inhibitor resistance (tumors published in Chaudhuri *et al*, Nature 2016, PMID: 27443740). In line with expectations, we did not find altered expression of *Fbxo5* (encoding Emi1). These data have now been added in Supplemental Figure 7D.

Comment 3: *“As they suggested in the end of the discussion WEE1 inhibitors could potentiate the effect of PARP inhibitors in BRCA2 mutant cells. Did they try to test this? It would reinforce the role of mitosis.”*

Reply: We are currently investigating the sensitivity of HR-deficient tumors for Wee1 inhibitors, in the presence or absence of PARP inhibitors. In line with our model, we found that Wee1 inhibition using AZD-1775 (formerly known as MK-1775) potentiates the effects of olaparib in DLD-1 cells. Of note, Wee1 inhibition already potently kills p53-mutant cancer cells, and in line with this notion, DLD1-wt *BRCA2* are also relatively sensitive to Wee1 inhibitors. These data have now been added in Supplemental Figure 8.

Comment 4: *“Figure 1E-F: The legend does not explain what the value between brackets is, I assume it is median or average but it should be explained.”*

Reply: We apologize for not indicating the meaning of the values indicated in Figure 1E-F. They represent medians, which has now been indicated in the figure legends.

Comment 5: *“Figure 2: instead of showing representative experiments they could show all the repetitions with standard deviation on the graph. It would be more convincing even though the results are really clear.”*

Reply: We have now included averages of the three experiments with error bars indicating standard deviations.

Comment 6: *“Figure 3: H, I and J legends are missing, again it might be better to show the repetitions on the experiments with standard deviation.”*

Reply: we apologize for missing legends. We have added the missing text, and have again shown the averages of all experiments with standards deviations.

Comment 7: *“Figure 4: In the text it is indicated Figures 4E-I and there is no I panel, the authors needs to correct this. In Figure 4G there is no statistical difference at 48hrs but there is now at 72hrs even though the differences are similar, there must be a problem here. I assume the standard deviations represent repetitions but is not written in the legend.”*

Reply: we have corrected the figure references and have corrected the legend text. Concerning the statistical difference in Figure 4G (now Figure 4E), the difference at 48 hours has a p-value of 0.0723, which we did not consider statistically significant, as we used a cut-off at p<0.05.

Comment 8: *“References 28 and 35 are the same”.*

Reply: thanks for indicating this error. We have now changed the reference list.

Reviewer #3

We thank the reviewer for the constructive comments, which we believe have contributed to strengthening of the manuscript.

Comment 1: *“There are existing papers in literature (not cited by the authors) reporting (i) that DNA replication stress-associated lesions are transmitted to S-phase, resulting in chromosomal mis-segregation and failed cytokinesis (Ichijima et al., Plos One, 2010) (ii) Cells deficient in Fanconi Anemia proteins (hence enhanced replication stress) display an increased number of inter-chromatid bridges resulting in failed cytokinesis and subsequent cell death via apoptosis (Vinciguerra et al, 2010, J Clin Invest) (iii) Loss of ZNF365, a protein involved in resolving replication stress at fragile sites leads to formation of anaphase bridges and cytokinesis failure (Zhang et al., 2013, Cancer Discovery). Based on the previous study (Vinciguerra et al., 2010, J Clin Invest), the claim made by the authors that “evidence that impaired replication directly results in cell death is lacking” is not true. Overall, it is already known that increasing replication stress, which in this manuscript was done by treating HR-deficient cells with PARPi, leads to the mis-segregation of chromosomes, failed cytokinesis and subsequent cell death.”*

Reply: We agree that it was previously reported that replication stress-induced DNA lesions can be transmitted to mitosis and can interfere with chromosome segregation, and have cited multiple papers showing this (Chan et al, Nature Cell Biology, 2009; Naim et al, Nature Cell Biology, 2013; Ying et al, Nature Cell Biology, 2013). We realize that additional literature underscores this notion, and we have added the indicated references (now included as references 28-30). The reviewer is correct that it is known that replication stress (which can be due to multiple defects) can be transmitted into mitosis, causes mitotic defects and is associated with cell death. Importantly, this phenomenon was not described for the clinically actionable scenario of PARP inhibition in BRCA1/2 defective cells, and mitosis was never causally related to cell death induction.

Comment 2: *“In Fig 1-3, olaparib is the only inhibitor used. Would similar results be expected with other PARP inhibitors and platinum agents? Is this unique to PARP inhibition or a general result of damage accumulation?”*

Reply: To address this relevant point, we have now included a second (and chemically unrelated) PARP inhibitor, AZD2461 (Figures 3C, 3D and Suppl. Figure 4B), and observed very similar results. To test whether PARP trapping is required for the observed phenotype, we also included PARP siRNA (see also comment 1 to reviewer #2, new Figure 3A, 3B and Suppl. Figure 4A), and found that indeed PARP trapping is required. Finally, we also included cisplatin treatment, to test whether the observed phenotype was generic for DNA damage accumulation. Surprisingly, cisplatin treatment also induced anaphase chromatin bridges, but to a much lesser extent when compared to PARP inhibition (now included in Figure 3C, 3D and Suppl. Figure 4B). This was not due to the absence of cisplatin-induced DNA lesions, as a clear increase in γ -H2AX was observed, (much higher than observed upon PARP inhibition, Figure 3C). Combined, these new data show that the observed PARP inhibitor-induced mitotic defects are not a generic response to DNA damage, require PARP trapping, and were confirmed using a second PARP inhibitor.

Comment 3: *“The authors suggest that aberrant replication intermediates caused by PARP inhibition persist into mitosis and that these intermediates are the cause of chromatin bridge formation and cytotoxicity. While this is plausible, PARP inhibition during mitosis could also directly affect the repair of DNA lesions encountered during this phase. To address these possibilities, cell cycle synchronizations with shorter PARP inhibitor treatments could be used.”*

Reply: To test whether PARP inhibitor requires S phase progression to induce chromatin bridges, we synchronized cells using a double thymidine block, and applied PARP inhibitor either during S-phase or at a time point when cells had progressed past S-phase. These data are presented in new Figures 3E, F and G. Clearly, treatment during S phase is required to efficiently induce mitotic FANCD2 foci (Figure 3F) as well as chromosome bridges and lagging chromosomes (Figure 3G). As a control, we treated cells at a time-point where cells has completed S phase (7 hours after release from a thymidine block). In these cells, PARP was inhibited when they entered mitosis, yet these cells did not display chromosome bridges and lagging chromosomes. These data strongly suggest that aberrant replication intermediates underlie the observed mitotic chromosome defects.

Comment 4: *“If the compromised fork stability leads to mitotic DNA damage and hence failed cytokinesis, then it is important to demonstrate that this mitotic DNA damage generated by PARPi can be rescued by protecting the replication fork by either inhibiting Mre-11 or knocking out Ptip1.”*

Reply: We have used PTIP depletion and Mre11 inhibition using Mirin to address this point. As shown in new Supplemental 1A and 1B, both PTIP depletion and Mre11 inhibition can rescue the replication fork destabilization induced by PARP inhibition in BRCA2-depleted cells. Importantly, either PTIP depletion or Mre11 inhibition partially rescues of the number of mitotic aberrancies, underscoring that replication fork stability indeed underlies the observed phenotype. These new data are discussed in the results section on page 5: “In line with.... ..fork protection defects (Suppl. Figures 1A, B).”, and on page 7/8: “Since the formation... .. not reduced upon Mre11 or PTIP inactivation, suggesting different biological origins of these lesion (Suppl. Figures 3F, G).”.

Comment 5: *“What is the fate of FANCD2+ cells resulting from PARP inhibitor treatment? Do the ones with high numbers of foci have a propensity to become multinucleated or die?”*

Reply: To address this point, we tried several ways to establish cells stably expressing GFP-FANCD2 in BRCA2-wt or BRCA2 mutant cell lines. Unfortunately, we were not able to reliably visualize and quantify GFP-FANCD2 foci in mitotic cells, which precluded follow-up analysis of these cells using live cell imaging. Combined analysis of endogenous FANCD2 foci during mitosis, in conjunction with analysis of chromosome bridges and lagging chromosomes, we found that cells with mitotic FANCD2 foci are more frequently found in cells with chromosome bridges (Suppl. Figure 4C).

Comment 6: *“Would the authors expect an increase in DSBs/gH2AX foci in interphase and mitotic cells after PARP inhibitor treatment? How would this compare to FANCD2 foci?”*

Reply: Based on the dogma that unrepaired single-stranded breaks (SSBs) lead to double stranded breaks (DSBs) during replication, one would also expect elevated levels of H2AX during interphase upon PARP inhibitor treatment. We have including

analysis of γ -H2AX in interphase (new Supplemental Figure 1C), and observed a statistically significant, yet modest, increase of γ -H2AX foci in interphase. When γ -H2AX foci were assessed in mitotic cells, a more pronounced increase was found (new Supplemental Figure 1D), suggesting that at least a subset of PARP-inhibitor-induced replication intermediates is transferred into mitosis, and only then may be processed or converted into DSBs.

Comment 7: *“Did cell death from olaparib in multinucleated cells occur during the subsequent interphase or during a subsequent mitosis? This type of cell death was frequent in HR-deficient cells treated with olaparib. If the multinucleation and accumulation of replication intermediates were directly detrimental to cells, one would expect EMI1 depleted cells to be sensitive to PARP inhibition. However, the lack of sensitivity may point to a direct role of PARP during mitotic repair. This could be addressed with experiments outlined in Comment 2.”*

Reply: We have visualized the behavior of individual cells in Supplemental Figure S5A. These plots illustrate that only very few cells die during the first mitosis (2 out of 215). The far majority of observed cell death occurs in the interphase after the first mitosis. In our analysis, we only imaged few cells that underwent two rounds of mitosis (3 out of 215). The observation that olaparib-treatment in BRCA2-depleted cells did not obviously delay mitotic duration, while mitotic bypass upon EMI1-depletion rescues survival suggests that the cue for cell death originates in mitosis, but requires additional time for execution. The experiments using cell cycle synchronization, in which treatment past S-phase -but during mitosis- lowered mitotic aberrancies, argue against a major role of PARP in mitotic repair.

Comment 8: *“Does PARP inhibition still lead to DNA damage (FANCD2 foci) in HR-deficient cells depleted for EMI1? On a similar note, can the authors use a method of mitotic inhibition that does not cause multinucleation? While this is a good control for allowing continued replication, the mechanism of multinucleation itself may confer some degree of resistance.”*

Reply: To address this point, we have analyzed γ -H2AX foci in EMI1-depleted cells and control-depleted cells (Supplemental Figures 7A, B). We found that EMI1-depletion does not alleviate the number of γ -H2AX nor FANCD2 foci when compared to controls (actually, we observed more γ -H2AX foci). This might be due to DNA replication in cells with increased nuclear content. Thus, it appears that DNA lesions instigate differential signaling in mitotic cells; perhaps exertion of force on DNA during mitosis leads to other types of DNA lesions that potently induce cell death.

In the second part of the question, the reviewer suggests testing the impact of mitotic inhibition without causing multinucleation. Important to note is that EMI1 depletion does not cause multinucleation, but rather caused endoreplication with a single large nucleus. Concerning other means of mitotic inhibition, there are multiple mechanisms described how cells can enter rounds of endoreplication, which mostly involve cytokinesis failure. In such a scenario, cells do enter and exit mitosis, and such approach does not allow us to test the contribution of mitotic progression per se.

Comment 9 (Minor point): *“The nomenclature SCR for “siRNA targeting control sequence” is confusing. It will be helpful to just use the commonly used term “control siRNA”.”*

Reply: we have adapted the text accordingly in the legends of the figures and supplemental figures.

Comment 10 (Minor point): *“Olaparib is misspelled in Fig 1E and Fig 1F”*

Reply: we apologize for this mistake and have corrected the text.

Comment 11 (Minor point): *“Fig2a: The DAPI signal is saturated and the tubulin signal is diffused in the cytoplasm rather than being concentrated in the spindle.”*

Reply: we have replaced the images with clearer examples.

Comment 12 (Minor point): *“Fig 3a-3c: The results presented are already known in the literature and hence they can be moved to the Supplementary section.”*

Reply: We have reorganized the figures, and Figure panels 3A-C are now in Supplemental Figure S2.

Comment 13 (Minor point): *“Figure 4e: 1.35% rather than 1,35%”*

Reply: we have adapted this issue.

Comment 14 (Minor point): *“Abbreviation: DSBs needs to be defined in the first paragraph rather than in the second paragraph.”*

Reply: in the revised version DSBs is now defined in the first paragraph.

Comment 15 (Minor point): *“Pg9. First paragraph: Define “wt” in text or use “wild-type”.*

Reply: We have now defined “wt” on page 7, line 25, as this is the first instance where wild-type is abbreviated.

REVIEWERS' COMMENTS:

Reviewer #1 (Remarks to the Author):

In this revised manuscript, the authors performed extra experiments and addressed most of the comments. While I appreciate its improvement, the study presents only limited conceptual advance over what is known in the field of DNA repair and cancer treatment.

Reviewer #2 (Remarks to the Author):

The authors made huge efforts to respond to the comments from myself as well as those of the two other reviewers. In particular they demonstrated that the synthetic lethality between BRCA2 and PARP is specific of PARP inhibitors and is not observed using siRNAs against PARP1. In addition the effect of WEE1 inhibitor is a promising approach to improve treatments based of PARP inhibition. In summary I think the paper is now even more solid (the first version was already very good) and I strongly support its publication in Nature Communications.

Point-by-point rebuttal to reviewer comments:

Reviewer #1 (Remarks to the Author):

Comment: *'In this revised manuscript, the authors performed extra experiments and addressed most of the comments. While I appreciate its improvement, the study presents only limited conceptual advance over what is known in the field of DNA repair and cancer treatment.'*

Reply: Our findings build on the previously reported observations that 1) replication-dependent DNA lesions were shown to be transmitted into mitosis, and 2) PARP inhibitors were shown to interfere with DNA replication.

Our data show that PARP inhibitor-mediated DNA lesions are also transmitted into mitosis. Moreover, we find that progression through mitosis in the presence of DNA lesions is instrumental to inducing cell death. We feel that these combined findings provide significant conceptual advance over the current understanding how PARP inhibitors cause cell death in HR-deficient cancer cells.

Reviewer #2 (Remarks to the Author):

Comment: *'The authors made huge efforts to respond to the comments from myself as well as those of the two other reviewers. In particular they demonstrated that the synthetic lethality between BRCA2 and PARP is specific of PARP inhibitors and is not observed using siRNAs against PARP1. In addition the effect of WEE1 inhibitor is a promising approach to improve treatments based of PARP inhibition. In summary I think the paper is now even more solid (the first version was already very good) and I strongly support its publication in Nature Communications.'*

Reply: We thank the reviewer for the comments.